# Epistemic Gain, Aleatoric Cost: Uncertainty Decomposition in Multi-Agent Debate for Math Reasoning

**Dan Qiao** [1 2 †]  **Binbin Chen** [2 §]  **Fengyu Cai** [3]  **Jianlong Chen** [1]  **Wenhao Li** [4]  **Fuxin Jiang** [2]  **Zuzhi Chen** [2]
**Hongyuan Zha** [1]  **Tieying Zhang** [2 ‡]  **Baoxiang Wang** [1 5 ‡]

## Abstract

Multi-Agent Debate (MAD) has shown promise in improving reasoning and reducing hallucinations, yet it remains unclear how information exchange shapes individual reasoning behavior. Empirically, MAD exhibits paradoxical phenomena, including rising accuracy with increasing token entropy and marked differences between homogeneous and heterogeneous agent combinations. In this paper, we introduce a Bayesian uncertainty analysis framework for MAD, which decomposes answer-level predictive uncertainty into epistemic uncertainty and aleatoric uncertainty, corresponding to the potential gain and cost of debate. Across multiple agent configurations, we find that effective debate depends on achieving high epistemic gain under controlled aleatoric cost. Building on this insight, we design an uncertainty-guided multi-agent reinforcement learning algorithm that encourages lower aleatoric cost and more effective epistemic information utilization. Experiments show that our approach simultaneously enhances each agent's accuracy and promotes a more productive debate process, providing an operational Bayesian perspective for understanding and improving MAD.

## 1. Introduction

Recent advances in Large Language Models (LLMs) have demonstrated remarkable general-purpose capabilities, yet these models remain susceptible to hallucinations and brittle reasoning in complex scenarios such as math problems. Inspired by human argumentation, Multi-Agent Debate (MAD) has emerged as a prominent inference-time paradigm to achieve more reliable responses by integrating perspectives from multiple source LLMs (Du et al., 2024). Specifically, MAD constructs input for each round by orchestrating task roles, prompt strategies, or communication topologies across multiple LLMs, and aggregates the final responses through majority voting or the LLM-as-a-Judge approach. Previous studies have shown that iterative debate can elicit a "wisdom of crowds" effect (Minsky, 1986), improving final-answer performance and often being attributed to verification and self-correction behaviors.

However, the underlying mechanism driving these performance gains remains largely opaque. While aggregated outcomes show improvement, the validity of the iterative debate process is increasingly being scrutinized. Choi et al. (2026) remove the majority voting process and find that the mean accuracy of debate among homogeneous agents often stagnates or even degrades across debate rounds, which implies that the effectiveness of MAD mainly stems from the aggregation process rather than belief improvement. Furthermore, Wynn et al. (2025) have highlighted a concerning frequency of correct answers flipping to incorrect ones during debate, suggesting that LLMs may be driven more by social conformity or sycophantic behavior (Sharma et al., 2024) than by logical deduction. Yao et al. (2025) have also pointed out that the inherent sycophancy of LLMs can hinder the development of positive disagreements into positive consensus during debates, which can prevent constructive disagreements from being effectively turned into improved reasoning. These findings indicate that the debate process may not be effective in reliable belief improvement and effective peer-information utilization, highlighting a central limitation of current MAD (Wang et al., 2024).

In this paper, to understand how information exchange among agents fundamentally shapes reasoning behavior in MAD, we focus on uncertainty quantification (Lakshminarayanan et al., 2017) of model predictive responses across debate turns and input contexts in math reasoning prob-

---

[†]Work done as an intern at ByteBrain, ByteDance. (Email: danqiao@link.cuhk.edu.cn). [§]Project Lead. [1]School of Data Science, The Chinese University of Hong Kong, Shenzhen, China [2]ByteDance, Beijing, China [3]Technical University of Darmstadt, Darmstadt, Germany [4]School of Computer Science, Tongji University, Shanghai, China [5]Vector Institute. Correspondence to: Baoxiang Wang <bxwang@cuhk.edu.cn>, Tieying Zhang <zhangtieying@bytedance.com>.

*Proceedings of the 43rd International Conference on Machine Learning*, Seoul, South Korea. PMLR 306, 2026. Copyright 2026 by the author(s).

lems. As shown in Fig. 1, accuracy rapidly increases and saturates as the debate progresses, accompanied by a noticeable increase in token entropy. This counterintuitive pattern suggests that debate induces non-trivial changes in model uncertainty. While token entropy provides an initial symptom of debate-induced instability, it does not distinguish whether uncertainty arises from disagreement across agents or instability within each agent's own responses. Motivated by this, we study MAD through a Bayesian uncertainty perspective and instantiate it with answer-level predictive distributions, where epistemic uncertainty captures cross-agent disagreement and the potential for information gain, while aleatoric uncertainty captures within-agent response instability introduced during reasoning.

In particular, we remove the final aggregation stage of MAD, including majority voting (Du et al., 2024) and weighted voting (Chen et al., 2024), to focus purely on the belief evolution of individual agents. This design allows us to separate improvements from answer aggregation from changes in individual agents' post-debate beliefs. It further enables us to move beyond simple accuracy metrics and systematically characterize the dynamics of information flow, particularly within heterogeneous agent setups, by analyzing answer-level frequency distributions and entropy throughout the debate trajectory. Our empirical results across homogeneous and heterogeneous MAD suggest that the effectiveness of a debate process is often influenced by two factors: sufficient initial inter-agent disagreement to create information-exchange potential and effective control of aleatoric uncertainty during the debate process. The former represents the potential for cognitive conflict and epistemic gain, while the latter reflects the instability introduced when agents incorporate external responses into their own reasoning. These observations suggest that inference-time prompting alone may be insufficient to reliably regulate the uncertainty dynamics of debate. To address this challenge, we present a multi-agent reinforcement learning (MARL) method that uses uncertainty-guided reward signals to improve external information utilization and reduce reasoning instability.

Our main contributions are summarized as follows:

- We propose a Bayesian uncertainty analysis framework for MAD, which decomposes answer-level predictive uncertainty into epistemic and aleatoric components through a computable Jensen-Shannon-based decomposition, providing an operational perspective for diagnosing debate dynamics.

- Through extensive evaluation of homogeneous and heterogeneous MAD across 7 base models, we characterize the uncertainty trade-off that constrains inference-time debate, showing that heterogeneous agents improve debate when their epistemic potential is realized under controlled aleatoric cost.

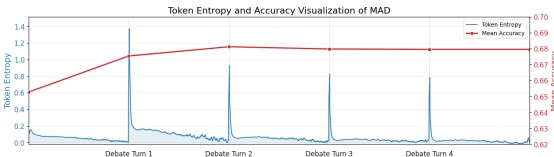

*Figure 1.* Multi agent debate performance with two `Qwen2.5-3B-Instruct` models. As accuracy (red line) increases across debate rounds, token entropy (blue shaded area) also increases noticeably.

- We propose UMAD[1], an uncertainty-guided MARL algorithm that combines advantage shaping with epistemic influence rewards to fine-tune agents. Experiments show that UMAD improves post-debate accuracy and peer information utilization, which achieves stronger post-debate performance than independently trained single-agent GRPO baselines. These results suggest that uncertainty-guided training can improve both reasoning ability and debating effectiveness, rather than just improving aggregation accuracy.

## 2. Problem Formulation

### 2.1. Multi-Agent Debate

Following standard MAD settings (Du et al., 2024; Park et al., 2025), we formulate MAD as a $T$-steps iterative generation process among $N$ agents, where the agents are parameterized by LLM parameters $\{\pi_{\theta_i}\}_{i=1}^{N}$. For a given problem $x \sim \mathcal{D}$ sampled from the dataset, there exists a ground truth answer as $y^*$. For math tasks, the reasoning process can be diverse, but the answer is usually numerically unique and can often be precisely verified through string matching against ground truth answer.[2] At each round $t \in \{0, \ldots, T\}$, agent $i$ generates a response $y_{i,t}$ conditioned on the original problem $x$ and a changing context $c_{i,t}$ as $y_{i,t} \sim \pi_{\theta_i}(\cdot \mid x, c_{i,t})$, where the context includes the reference solutions from other agents and additional debate instructions. In the initial round $t = 0$, the models usually answer the original question $x$ directly, where the auxiliary context $c_{i,0}$ is empty. Formally, this response $y_{i,t}$ with $L$ tokens is generated autoregressively by agent $i$ as follows:

$$\pi_{\theta_i}(y_{i,t} \mid x, c_{i,t}) = \prod_{l=1}^{L} \pi_{\theta_i}(y_l \mid x, c_{i,t}, y_{<l}).$$

The response usually contains intermediate reasoning steps before generating the final answer, which may be enclosed with special symbols `</think>` in the reasoning models

---

[1]Our source code is available at `https://github.com/qiaodan-cuhk/UMAD`.

[2]With a slight abuse of notation, we use $y$ to represent both the full responses and the extracted final answers throughout the paper.

such as OpenAI o1 and DeepSeek-R1 (OpenAI; Guo et al., 2025). Although there are various information schemes for MAD prompts (Li et al., 2024; Liu et al., 2024; Chen et al., 2024), we define a uniform context mapping function $\Phi$ that constructs the input context for each agent based on the global history of peer responses from the previous rounds as $c_{i,t} = \Phi_i(\{Y_k\}_{k=1}^{t-1}, \mathcal{G}, \mathcal{P})$, where $Y_k = \{y_{1,k}, \ldots, y_{N,k}\}$ represents all agents' responses in debate turn $k$, communication topology $\mathcal{G}$ determines the range of peer agents responses that agent $i$ can access (e.g., fully connected graphs and directed graphs). Agent role prompts $\mathcal{P}$ determine how each agent utilizes and processes peer agents responses, such as concatenation, introducing additional LLM for summarization, and specific role instructions. At the final turn $T$, the system prediction results $\hat{y} = \Psi(Y_T)$ will be distilled from all agents' responses $Y_T$ by an aggregation function $\Psi$, such as majority voting (Du et al., 2024), Elo (Zhang & Xiong, 2025), and LLM-as-a-Judge (Zhu et al., 2025).

In this work, we focus on individual agents' reasoning capabilities rather than optimizing the MAD workflows for final responses accuracy. We therefore adopt a fully connected communication graph $\mathcal{G}$, where each agent observes all peer responses from the previous round, and do not introduce specialized roles, summarization modules, or task-specific communication rules. We also remove the final aggregation function $\Psi$ in our experiments and analysis, so that evaluation directly reflects the reasoning ability of each agent's post-debate response $y_{i,T}$. This setting isolates the effect of information exchange on individual reasoning behavior.

### 2.2. Reinforcement Learning with Verifiable Rewards

Reinforcement Learning with Verifiable Rewards (RLVR) has emerged as a powerful paradigm, which leverages rule-based verifiable binary sparse reward signals to stimulate the high-order complex reasoning capabilities of LLMs by RL training. Recent advances builds upon Group Relative Policy Optimization (GRPO) (Shao et al., 2024), a specialized RL algorithm for reasoning tasks that eliminates the need for training a critic model. In the standard single-agent setting, for each query $x$, the LLM policy $\pi_\theta$ samples a group of $G$ responses $\{y_1, \ldots, y_G\}$. For each response $y_i$, a reward function $R(y_i, y^*)$ evaluates the final answer and provide the binary rewards, which is $R = 1$ if the answer correct and $R = 0$ otherwise.

The GRPO objective is defined as:

$$\mathcal{J}(\theta) = \mathbb{E}_{x \sim \mathcal{D}, \{y_i\}_{i=1}^G \sim \pi_\theta(\cdot|x)} \Big[ \frac{1}{G} \sum_{i=1}^G \frac{1}{|y_i|} \sum_{j=1}^{|y_i|} \Big( \min\Big(r_{i,j}(\theta) A_{i,j},$$

$$\mathrm{clip}(r_{i,j}, 1-\epsilon, 1+\epsilon) A_{i,j}\Big) - \beta D_{\mathrm{KL}}(\pi_\theta \| \pi_{\mathrm{ref}})\Big)\Big], \quad (1)$$

where $r_{i,j} = \frac{\pi_\theta(y_{i,j}|x)}{\pi_{\theta_{old}}(y_{i,j}|x)}$ represents the token-level ratio,

$A_{i,j} = \frac{R_i - \mathrm{mean}(\{R\}_{i=1}^G)}{\mathrm{std}(\{R\}_{i=1}^G)}$ represents the group-normalized advantage, and $\beta$ controls the KL divergence penalty. This critic-free method can effectively stabilize the training of LLM inference and is widely used in math, code, and agentic training tasks (Yu et al., 2026; Wei et al., 2026; Jin et al., 2025). In our work, we extend GRPO to multi-agent debate by treating MAD as a learnable multi-agent, multi-step decision process and optimizing each policy $\pi_{\theta_i}$ with round-level verifiable rewards, as described in Section 4.1.

## 3. Uncertainty Mechanism in MAD

In this section, we analyze the dynamics of multi-agent debate (MAD) through an uncertainty decomposition lens. We treat the Bayesian formulation as a conceptual framework and explicitly distinguish latent belief variables, answer-level diagnostic proxies, and training-time optimization proxies. Section 3.1 formalizes debate as an iterative Bayesian belief update process in which peer responses act as noisy evidence. Section 3.2 instantiates this view with answer-level predictive distributions and decomposes system-level uncertainty into epistemic and aleatoric components. Section 3.3 further studies individual-agent uncertainty under changing debate contexts and explains when heterogeneous peers can provide additional epistemic gain.

### 3.1. Iterative Bayesian Belief Update of MAD

Motivated by recent research about LLMs as implicit Bayesian inference process (Jayasekera et al., 2025; Xie et al., 2022; Choi et al., 2026), we formalize agent $i$'s response generation as a hierarchical sampling process to isolate the latent beliefs $\varphi_{i,t}$ from the observed text responses $y_{i,t}$. The latent belief variable $\varphi_{i,t}$ denotes the agent's implicit distribution over solution hypotheses, including relevant theorems, intermediate reasoning plans, and candidate final answers given the debate context $c_{i,t}$[3]. This can reflect the agent's current understanding of the problem, such as mathematical theorems, analytical methods, and problem-solving logic. The observed text response $y_{i,t}$ is then obtained by autoregressive sampling in the vocabulary space by the LLM, which can be considered as being driven by the latent belief $\varphi_i$ as $y_{i,t} \sim p(y \mid \varphi_i)$. Since the LLM parameters $\theta_i$ is frozen in the inference time of MAD, the posterior predictive distribution over responses is obtained by marginalizing out the latent belief as

$$p(y_{i,t} \mid x, c_{i,t}) = \int p(y_{i,t} \mid \varphi_i) \, p(\varphi_i \mid x, c_{i,t}) \, d\varphi_i.$$

Therefore, the process of updating responses through multiple rounds of debate in MAD can be conceptually understood as an iterative Bayesian belief update process. By

---

[3]For notational simplicity, the turn index $t$ in $\varphi_{i,t}$ is omitted hereafter when clear from the context.

adding additional reference solutions to change the context $c_{i,t}$ between rounds, the agent's latent belief $p(\varphi_i \mid x, c_{i,t})$ is implicitly updated, which in turn changes the posterior prediction distribution $p(y_{i,t} \mid x, c_{i,t})$ and may change the final answer. For example, compared with the first turn initial responses $y_{i,0}$, the updated belief at turn $t$ can be modeled as the Bayesian update over the latent variable as

$$p(\varphi_i \mid x, c_{i,t}) \propto p(c_{i,t} \mid \varphi_i, x) \, p(\varphi_i \mid x), \qquad (2)$$

where $p(c_{i,t} \mid \varphi_i, x)$ represents the implicit likelihood of the peer agents' reference solutions given the agent's latent belief state under context $c_{i,t}$. Intuitively, the effective debate contexts can shift the agent's posterior belief towards the ground truth and thus improve the predictive accuracy with better posterior $p(y_{i,t} \mid x, c_{i,t})$.

Unlike multiple-choice questions with limited options, math reasoning problems may often have an infinite space of answers. Considering that the ground-truth answer is usually unique up to an equivalence relation, we can simplify the belief space with binary correctness event for agent $i$ at turn $t$ as $h_{i,t} := \mathbf{1}[y_{i,t} \in \mathcal{Y}^*] \in \{0, 1\}$, where $\mathcal{Y}^* \subseteq \mathcal{Y}$ denotes the set of responses equivalent to the ground-truth answer. Accordingly, the predictive belief of correctness is

$$p(h_{i,t} = 1 \mid x, c_{i,t}) = \sum_{y \in \mathcal{Y}^*} p(y_{i,t} = y \mid x, c_{i,t}).$$

For notational simplicity, we omit the agent index and write $c_t$ for the debate context received by a fixed agent. Then we have the following lemma regarding the correctness of agent $i$'s responses at turn $t$ as:

**Lemma 3.1** (Noisy-evidence Log-odds Update)**.** *Given an input question $x$, assuming that the initial context is empty and debate context $c_t$ is valid as $p(c_t \mid x) > 0$, the log-odds of the binary hypothesis $h \in \{0, 1\}$ after debate can be written as*

$$\log \frac{p(h = 1 \mid x, c_t)}{p(h = 0 \mid x, c_t)} = \log \frac{p(h = 1 \mid x)}{p(h = 0 \mid x)} + \log \frac{p(c_t \mid h = 1, x)}{p(c_t \mid h = 0, x)}.$$

The proof of Lemma 3.1 follows directly from Bayes' rule and is provided in Appendix B.1.

The additive term $\log \frac{p(c_t \mid h=1, x)}{p(c_t \mid h=0, x)}$ is the *log Bayes factor* induced by the debate context. A positive value indicates that the context provides evidence in favor of correctness and may induce a wrong-to-correct ($W \to C$) transition. A negative value indicates misleading or poorly utilized evidence, which can shift the posterior belief away from the ground truth and induce a correct-to-wrong ($C \to W$) transition. It provides a Bayesian interpretation of debate stagnation and answer flipping observed in prior MAD studies (Choi et al., 2026; Wynn et al., 2025; Yao et al., 2025). This observation highlights that MAD does not guarantee monotonic utilization of correct peer cues: even when peer solutions contain useful information, an agent's internal belief can still be shifted toward an incorrect response.

## 3.2. System-Level Uncertainty Trade-off in MAD

Quantifying the uncertainty of large models is crucial for understanding the reliability of the responses. However, due to the large scale of LLM parameters and high-dimensional generation space, traditional uncertainty estimation methods based on deep Bayesian networks or dropout inference (Lakshminarayanan et al., 2017; Li & Gal, 2017) are impractical for LLM-based reasoning. Moreover, the implicit latent belief state $\varphi_i$ is also computationally intractable for large LLMs (Abbasi Yadkori et al., 2024; Jayasekera et al., 2025). To address the above limitations, we consider quantifying and decomposing the uncertainty of the MAD system based on the answer-level predictive distributions.

The core intuition is that the MAD system can be viewed as an ensemble of predictive experts. Given the same question $x$ and debate contexts $C_t = \{c_{i,t}\}_{i=1}^N$, each agent induces an answer-level predictive distribution $p_{i,t}(y)$. In practice, we estimate $p_{i,t}(y)$ by sampling $G$ responses from agent $i$ under the same $(x, c_{i,t})$ and extracting their final answers. This empirical distribution captures how stable or dispersed the agent's final-answer predictions are under the current debate context. Instead of treating these samples as exact posterior draws from a joint Bayesian model, we use an ensemble-of-experts proxy and define the system-level predictive distribution as $p_{\text{sys},t} := \frac{1}{N} \sum_{i=1}^N p_{i,t}$.

This ensemble proxy also aligns with Bayesian belief update under the mild assumption that each agent's distribution $p_i(y)$ can be viewed as an expert corresponding to implicit sampling, such that the cross-expert average approximates the Bayesian model average. Therefore, the mixture distribution $p_{\text{sys},t}$ can be interpreted as an approximate posterior prediction distribution. We quantify total uncertainty (TU) as the entropy of the collective answer-level predictive distribution. This mixture entropy admits an answer-level analogue of the classical law of total entropy (Lakshminarayanan et al., 2017), which can be decomposed into system epistemic uncertainty (Sys-EU) and system aleatoric uncertainty (Sys-AU) as follows.

**Proposition 3.2** (Answer-level System Uncertainty Decomposition)**.** *Let $\{p_{i,t}\}_{i=1}^N$ denote the agents' answer-level predictive distributions at turn $t$, and let $p_{\text{sys},t} = \frac{1}{N} \sum_{i=1}^N p_{i,t}$ be their mixture. Then the answer-level total uncertainty decomposes as*

$$\underbrace{H(p_{\text{sys},t})}_{\text{TU}(t)} = \underbrace{\text{JSD}(p_{1,t}, \ldots, p_{N,t})}_{\text{Sys-EU}} + \underbrace{\frac{1}{N} \sum_{i=1}^N H(p_{i,t})}_{\text{Sys-AU}}. \quad (3)$$

The proof of Proposition 3.2 follows from the definition of generalized Jensen-Shannon divergence and is provided in Appendix B.2. This identity is exact for the answer-level mixture distribution, while its interpretation as epistemic

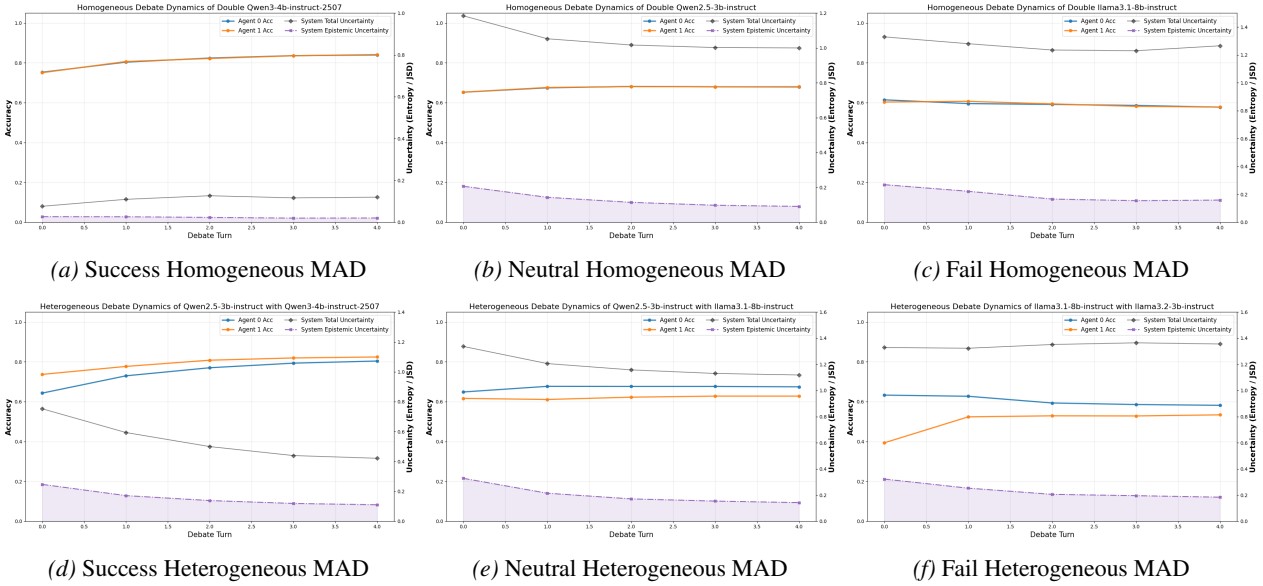

*Figure 2.* Overview of uncertainty decomposition in MAD. The purple dashed area represents the system epistemic uncertainty and the gray line represents the total uncertainty. The gap between total uncertainty and Sys-EU corresponds to Sys-AU. The blue and orange lines represent the accuracy on the test datasets. **Top row:** Homogeneous multi-agent debate. **Bottom row:** Heterogeneous multi-agent debate.

and aleatoric uncertainty is operational. Specifically, Sys-EU measures disagreement among agents' answer distributions and therefore serves as a proxy for reducible inter-agent uncertainty, while Sys-AU measures the average entropy within individual agents' answer distributions and serves as a proxy for response instability. This decomposition does not assume that Sys-EU and Sys-AU are statistically independent across debate turns; rather, it separates two additive components of answer-level entropy. In deterministic math reasoning tasks, Sys-AU does not mainly arise from inherent label ambiguity, but from generation stochasticity, long-context perturbations, and noisy utilization of peer responses (Feng et al., 2025; Ling et al., 2024).

To examine whether this operational decomposition captures debate dynamics, we evaluate both homogeneous and heterogeneous MAD configurations on a subset of the MATH (Hendrycks et al., 2021) dataset with $N = 2$ agents, $T = 5$ debate rounds, and $G = 16$ rollouts for every prompt (see Appendix D.5 for detailed setups). As presented in Fig. 2, across all settings, Sys-EU generally decreases with debate turns, suggesting that MAD tends to reduce inter-agent disagreement and drive agents toward consensus, although this consensus is not necessarily correct.

Crucially, performance differences across configurations are primarily explained by the balance between reducing Sys-EU and controlling Sys-AU. Successful debates keep Sys-AU low or decreasing, whereas failure cases exhibit persistent or increasing Sys-AU that offsets the benefit of disagreement reduction. This trade-off also clarifies when agent heterogeneity is advantageous. Heterogeneous pairs

typically exhibit higher initial Sys-EU than homogeneous counterparts, indicating a larger cognitive gap and greater potential for information exchange. However, this epistemic potential translates into gains only when the receiving agent can reliably process peer arguments without introducing excessive generation instability. Our flip-ratio analysis in Appendix D.4 supports this view: cross-family pairings can create large epistemic gaps but may also introduce additional instability, whereas within-family size or generation heterogeneity often offers a better balance. In practice, a stronger model with a low flip ratio can function as a stable anchor, while the weaker counterpart must retain sufficient instruction-following capacity to synthesize peer reasoning rather than oscillating or blindly conforming.

### 3.3. Epistemic Advantage of Heterogeneous MAD

In the previous section 3.2, the system-level uncertainty analysis shows that debate accuracy is jointly affected by epistemic gain and aleatoric cost, yet it is insufficient to explain when a specific peer response provides useful information to an individual agent. Therefore, we analyze the uncertainty change of a single agent when receiving homogeneous and heterogeneous peer messages. The key distinction is between mere disagreement and useful disagreement: high system-level disagreement indicates available epistemic potential, but an individual agent can benefit only when the peer message provides non-redundant evidence about its latent solution belief.

Following the Bayesian view in Section 3.1, let $\varphi$ denote the agent's latent belief over solution hypotheses under the

current problem $x$ and context $c$. We define the epistemic gain provided by an additional peer message $m$ as

$$G_{\text{epi}}(m \mid x, c) := I(\varphi; m \mid x, c).$$

This quantity measures how much information the peer message provides about the agent's latent belief state. In MAD, high epistemic gain corresponds to evidence that introduces useful solution hypotheses, corrects misleading reasoning paths, or helps the agent distinguish between competing answers. Operationally, repeated samples from an agent approximate the answer support reachable under its current belief, similar to a `pass@K`-style view. A peer is useful when its sampled responses cover solution modes that the receiving agent does not already cover. We next formalize why homogeneous debate often provides limited individual improvement. Homogeneous peers may generate different responses due to sampling randomness, but their answer supports and reasoning biases are often substantially overlapping. Thus, after observing one homogeneous message, another homogeneous message tends to provide only limited additional information.

**Assumption 3.3** (Homogeneous Information Saturation). Given the current problem $x$ and context $c$, let $m_{\text{homo}}$ and $m'_{\text{homo}}$ be two independent messages from homogeneous peers. After conditioning on one homogeneous message, the additional conditional information provided by another homogeneous message is bounded:

$$I(\varphi; m'_{\text{homo}} \mid x, c, m_{\text{homo}}) \leq \epsilon_{\text{homo}}.$$

This assumption is not a universal claim about all agents, but a diagnostic condition for the common case in which same-model samples have diminishing marginal novelty. However, it captures the support-overlap intuition: homogeneous agents may disagree at the surface level, but they often explore similar regions of the solution space. As a result, homogeneous debate can reduce disagreement and move agents toward consensus, but they may not substantially expand recipients' support for effective solutions (Choi et al., 2026; Zhang et al., 2025; Wynn et al., 2025).

**Theorem 3.4** (Heterogeneous epistemic advantage under complementarity). *Suppose Assumption 3.3 holds. Let $m_{\text{hetero}}$ be a message from a heterogeneous peer. If the heterogeneous message provides complementary evidence beyond the homogeneous saturation level, i.e.,*

$$I(\varphi; m_{\text{hetero}} \mid x, c, m_{\text{homo}}) \geq \epsilon_{\text{homo}} + \Delta$$

*for some $\Delta \geq 0$, then replacing the second homogeneous message with the heterogeneous message increases the total epistemic information by at least $\Delta$:*

$$I(\varphi; m_{\text{homo}}, m_{\text{hetero}} \mid x, c) - I(\varphi; m_{\text{homo}}, m'_{\text{homo}} \mid x, c) \geq \Delta.$$

The proof of Theorem 3.4 follows from the chain rule of conditional mutual information and is provided in Appendix B.3. It clarifies that the advantage of heterogeneous MAD comes from conditional novelty rather than model diversity itself. Heterogeneous agents can provide complementary evidence beyond the inference scope of homogeneous peers, expanding the receiver's effective solution support and creating greater wrong-to-correct potential. This also explains why homogeneous debate often stagnates: when the conditional novelty of another same-model message is small, disagreement reduction mainly reflects redundant consensus formation rather than genuine belief expansion (Wynn et al., 2025).

However, the theorem characterizes epistemic potential rather than guaranteed accuracy improvement. As shown in Section 3.2, this potential translates into performance gains only when the receiving agent can absorb complementary evidence under controlled aleatoric cost. This also explains the failure patterns in heterogeneous MAD: for example, the Llama-based failure cases in Fig. 2 still exhibit non-negligible disagreement reduction, but the induced response instability keeps Sys-AU high enough to offset the epistemic benefit. Therefore, the goal is not merely to maximize complementary epistemic gain $\Delta$, but to realize this gain while controlling aleatoric cost. This trade-off motivates our downstream *training-time optimization proxies*, which encourage agents to absorb useful heterogeneous evidence while suppressing instability during debate.

## 4. Uncertainty-Guided MARL for MAD

Based on the analysis in Section 3, effective MAD requires realizing complementary epistemic information from peers while controlling the aleatoric instability introduced during debate. Therefore, we model MAD as a MARL problem to improve how agents use peer information during debate. Our method, named **Uncertainty-Guided Multi-Agent Debate (UMAD)** Reinforcement Learning, optimizes the agent's policy using verifiable rewards of responses, encouraging agents to absorb useful peer evidence while suppressing unstable generations.

### 4.1. MAD as a Dec-POMDP

Motivated by recent works on cooperative training among multiple LLMs (Park et al., 2025; Liao et al., 2025; Wan et al., 2026), we formulate the multi-agent debate reinforcement learning as a decentralized partially observable Markov decision process (Dec-POMDP) (Oliehoek et al., 2016) defined by the tuple $\langle \mathcal{N}, \mathcal{S}, \mathcal{A}, P, R, \Omega, \mathcal{O}, \gamma \rangle$.

Specifically, at each round $t$, each agent $i \in \{1, \ldots, N\}$ observes a local context $o_{i,t} = x \oplus c_{i,t}$, consisting of the original query and the debate context constructed from pre-

vious peer responses. The response sequence generated by LLM $i$, denoted by $a_{i,t}$, is agent $i$'s action at step $t$ and contains intermediate reasoning steps and the final answer. To induce collaborative improvement, we employ a dense team reward $R_t = \sum_i r(a_{i,t}, y^*)$, where each individual reward is computed by verifying the extracted final answer. Considering the complexity and high computational requirements of training LLMs, we adopt an independent learning paradigm as the algorithm backbone, where each agent maximizes the expected joint return. However, standard independent learning often suffers from non-stationarity and coarse credit assignment in MAD. We address these using a two-pronged uncertainty-guided mechanism.

### 4.2. Uncertainty-Guided Optimization Mechanisms

Based on the uncertainty trade-off identified in Section 3, we introduce two tractable training-time proxies that encourage epistemic information utilization and reduce aleatoric instability during policy optimization.

**Aleatoric Uncertainty-Aware Advantage.** Agents in debate often exhibit miscalibrated confidence, manifesting as "stubborn hallucinations" or "hesitant reasoning." To regulate aleatoric noise, we employ a confidence-aware advantage shaping mechanism. Motivated by UCAS (Xie et al., 2025), we use token-level negative log-probability as a practical proxy for generation uncertainty. For each response $y_{i,t}$, we first compute the token mean negative log-probability as

$$\hat{\mathcal{U}}_{i,t} = -\frac{1}{|y_{i,t}|} \sum_{j=1}^{|y_{i,t}|} \log \pi_{\theta_i}(y_{i,t,j} \mid x, c_{i,t}, y_{i,t,<j}),$$

which can be standardize within the GRPO sampling group as $\bar{\mathcal{U}}_{i,t} = \frac{\hat{\mathcal{U}}_{i,t} - \mu_{\mathcal{G}}}{\sigma_{\mathcal{G}} + \epsilon}$. Then we modulate the standard advantage $A_{i,t}$ as

$$A_{i,t}^{\text{au}} = W(\bar{\mathcal{U}}_{i,t}) A_{i,t}, \qquad (4)$$

where the uncertainty-dependent weight is defined as $W(\bar{\mathcal{U}}_{i,t}) = \exp(-\alpha_{\text{au}} \bar{\mathcal{U}}_{i,t})$,, and $\alpha_{\text{au}}$ is a hyperparameter. This token-level signal is a practical proxy for response instability rather than an exact estimate of semantic aleatoric uncertainty. This shaping down-weights high-uncertainty responses, reducing the impact of noisy generations caused by long-context perturbations or heterogeneous peer contexts.

**Epistemic Influence Intrinsic Reward.** To encourage epistemic information gain $G_{\text{epi}}$, effective debate requires agents to be not only correct, but also useful to their peers. The core intuition is that the same peer message can have different effects on agents with different latent beliefs. Therefore, in addition to correctness rewards, we need to incentivize responses that can be more effectively utilized by others. We introduce an intrinsic reward $r_{i,t}^{\text{eu}}$ that quantifies the ob-

servable positive influence on peers:

$$r_{i,t}^{\text{eu}} = \frac{\eta}{N-1} \sum_{j \neq i} \Delta R_{\text{peer}}(j, t+1), \qquad (5)$$

where $\Delta R_{\text{peer}}$ denotes the average correctness improvement of multiple rollout trajectories in GRPO with agent $i$'s reference solutions. While this mechanism does not directly compute mutual information, it serves as an observable proxy for epistemic influence by rewarding responses that improve peers' next-round average correctness.

## 5. Experiments

### 5.1. Experiment Settings

**Debate Configurations.** For homogeneous debate, we use two `Qwen2.5-3B-Instruct` agents as $A0$ and $A1$, which provide a stable same-model setting with reliable instruction following and math reasoning ability. To construct a heterogeneous debate system with model diversity while keeping instruction-following ability comparable, we select two different models with generational differences from the Qwen series as `Qwen2.5-3B-Instruct` ($A0$) and `Qwen3-4B-Instruct-2507` ($A1$) (Yang et al., 2024a; 2025a).

**Datasets.** The agents are trained on the MATH dataset (Hendrycks et al., 2021) with 7,500 training samples using a two-round-debate training protocol with $T_{train} = 2$. This setup evaluates whether a debate policy learned from short local interactions can generalize to longer inference-time debates rather than requiring expensive long-horizon training. During inference, we extend the debate to $T_{test} = 5$ rounds to evaluate robustness and generalization against mode collapse. We assess in-distribution performance on MATH500 dataset with 500 samples from MATH and generalization capability on out-of-distribution tasks ranging from grade-school math (GSM8K (Cobbe et al., 2021)) to olympiad-level competitions (AMC2023, AIME24, AIME25 (Knovel Engineering, 2025; Art of Problem Solving, 2024; 2025)).

**Baselines.** We compare UMAD with Zero-Shot MAD and Standard IPPO in the main table. Zero-Shot MAD serves as the inference-time debate baseline, while Standard IPPO is a multi-agent RL baseline without uncertainty guidance. In Appendix E.1, we further include a strong Single-GRPO Pair baseline, where each model is independently fine-tuned with standard GRPO settings for 15 epochs until convergence and then evaluated under the same debate protocol. This baseline uses more standalone training than UMAD and tests whether gains come merely from stronger individual models. Unless otherwise specified, all main results are averaged over five evaluation seeds, and the reported $\pm$ values indicate evaluation variability across seeds.

*Table 1.* Comparisons of individual agent accuracy across initial responses ($T = 1$) and debate responses ($T = 2, 5$). We report `pass@1` results for both homogeneous and heterogeneous MAD settings using `Qwen2.5-3B-Instruct` and `Qwen3-4B-Instruct-2507`. Light blue + Gain and light pink - Loss indicate performance variations. Cell colors are assigned relative to the corresponding `Zero-Shot` baseline at the same turn ($T$). Bold values indicate the best performance among `Zero-Shot`, `IPPO`, and `UMAD` at $T = 5$.

| Setting | Method (Round) | GSM8K | | MATH500 | | AMC23 | | AIME24 | | AIME25 | | Average | |
|---|---|---|---|---|---|---|---|---|---|---|---|---|---|
| | | A0 | A1 | A0 | A1 | A0 | A1 | A0 | A1 | A0 | A1 | A0 | A1 |
| Homogeneous MAD | Zero-Shot ($T=1$) | $83.3_{\pm0.4}$ | $83.3_{\pm0.4}$ | $61.0_{\pm0.5}$ | $61.0_{\pm0.5}$ | $35.0_{\pm1.2}$ | $45.0_{\pm1.1}$ | $3.3_{\pm0.0}$ | $0.0_{\pm0.0}$ | $2.0_{\pm1.8}$ | $2.0_{\pm1.8}$ | $36.9_{\pm0.4}$ | $38.3_{\pm0.4}$ |
| | IPPO ($T=1$) | $83.4_{\pm0.3}$ | $82.3_{\pm0.4}$ | $65.2_{\pm0.5}$ | $62.2_{\pm0.6}$ | $45.0_{\pm1.2}$ | $45.0_{\pm1.1}$ | $3.3_{\pm0.4}$ | $1.5_{\pm0.3}$ | $2.7_{\pm0.4}$ | $2.7_{\pm0.4}$ | $39.9_{\pm0.4}$ | $38.7_{\pm0.5}$ |
| | UMAD ($T=1$) | $82.6_{\pm0.4}$ | $83.7_{\pm0.3}$ | $62.4_{\pm0.5}$ | $63.4_{\pm0.4}$ | $42.5_{\pm1.4}$ | $37.5_{\pm1.6}$ | $4.0_{\pm0.5}$ | $2.0_{\pm0.4}$ | $3.3_{\pm0.6}$ | $3.3_{\pm0.5}$ | $39.0_{\pm0.5}$ | $38.0_{\pm0.6}$ |
| | Zero-Shot ($T=2$) | $83.4_{\pm0.3}$ | $84.1_{\pm0.4}$ | $63.0_{\pm0.5}$ | $64.6_{\pm0.6}$ | $30.0_{\pm1.3}$ | $40.0_{\pm1.2}$ | $3.3_{\pm0.0}$ | $2.0_{\pm1.8}$ | $6.7_{\pm0.0}$ | $2.0_{\pm1.8}$ | $37.3_{\pm0.0}$ | $38.5_{\pm0.6}$ |
| | IPPO ($T=2$) | $83.4_{\pm0.2}$ | $83.3_{\pm0.3}$ | $65.8_{\pm0.4}$ | $64.2_{\pm0.5}$ | $42.5_{\pm1.3}$ | $47.5_{\pm1.1}$ | $4.0_{\pm0.6}$ | $2.7_{\pm0.4}$ | $7.3_{\pm0.5}$ | $3.3_{\pm0.4}$ | $40.6_{\pm0.4}$ | $40.2_{\pm0.4}$ |
| | UMAD ($T=2$) | $83.9_{\pm0.3}$ | $84.1_{\pm0.2}$ | $63.4_{\pm0.4}$ | $63.8_{\pm0.5}$ | $47.5_{\pm1.2}$ | $50.0_{\pm1.0}$ | $4.7_{\pm0.7}$ | $3.3_{\pm0.5}$ | $8.0_{\pm0.8}$ | $4.0_{\pm0.6}$ | $41.5_{\pm0.4}$ | $41.0_{\pm0.5}$ |
| | Zero-Shot ($T=5$) | $83.4_{\pm0.4}$ | $\mathbf{84.6}_{\pm0.3}$ | $64.4_{\pm0.5}$ | $64.6_{\pm0.5}$ | $42.5_{\pm1.2}$ | $42.5_{\pm1.1}$ | $3.3_{\pm3.3}$ | $4.7_{\pm5.1}$ | $5.3_{\pm1.8}$ | $6.7_{\pm3.3}$ | $39.8_{\pm0.8}$ | $40.6_{\pm1.2}$ |
| | IPPO ($T=5$) | $83.5_{\pm0.3}$ | $83.7_{\pm0.2}$ | $\mathbf{65.8}_{\pm0.4}$ | $\mathbf{65.6}_{\pm0.4}$ | $\mathbf{47.5}_{\pm1.1}$ | $47.5_{\pm1.2}$ | $4.7_{\pm0.6}$ | $5.3_{\pm0.5}$ | $6.7_{\pm0.7}$ | $7.3_{\pm0.6}$ | $41.6_{\pm0.5}$ | $41.9_{\pm0.4}$ |
| | UMAD ($T=5$) | $\mathbf{83.9}_{\pm0.2}$ | $83.7_{\pm0.3}$ | $65.4_{\pm0.5}$ | $64.4_{\pm0.6}$ | $\mathbf{47.5}_{\pm1.3}$ | $\mathbf{52.5}_{\pm1.1}$ | $\mathbf{5.3}_{\pm0.7}$ | $\mathbf{6.0}_{\pm0.6}$ | $\mathbf{8.0}_{\pm0.9}$ | $\mathbf{8.7}_{\pm0.8}$ | $\mathbf{42.0}_{\pm0.4}$ | $\mathbf{43.1}_{\pm0.5}$ |
| Heterogeneous MAD | Zero-Shot ($T=1$) | $82.8_{\pm0.5}$ | $90.4_{\pm0.3}$ | $63.6_{\pm0.6}$ | $75.2_{\pm0.5}$ | $37.5_{\pm1.5}$ | $60.0_{\pm1.4}$ | $5.0_{\pm7.1}$ | $20.0_{\pm0.0}$ | $5.0_{\pm2.4}$ | $30.0_{\pm4.7}$ | $38.8_{\pm1.9}$ | $55.1_{\pm0.9}$ |
| | IPPO ($T=1$) | $82.0_{\pm0.4}$ | $90.1_{\pm0.3}$ | $62.6_{\pm0.5}$ | $76.0_{\pm0.6}$ | $27.5_{\pm4.2}$ | $55.0_{\pm3.8}$ | $4.2_{\pm1.5}$ | $18.9_{\pm2.1}$ | $4.2_{\pm1.3}$ | $28.9_{\pm2.8}$ | $36.1_{\pm1.5}$ | $53.8_{\pm1.8}$ |
| | UMAD ($T=1$) | $83.2_{\pm0.9}$ | $89.5_{\pm0.7}$ | $63.0_{\pm0.7}$ | $79.6_{\pm0.9}$ | $38.3_{\pm5.2}$ | $55.8_{\pm3.8}$ | $3.3_{\pm3.3}$ | $17.8_{\pm3.8}$ | $0.0_{\pm0.0}$ | $27.8_{\pm8.4}$ | $37.6_{\pm0.6}$ | $54.1_{\pm2.1}$ |
| | Zero-Shot ($T=2$) | $85.6_{\pm0.4}$ | $90.3_{\pm0.4}$ | $72.0_{\pm0.6}$ | $77.8_{\pm0.5}$ | $50.0_{\pm1.6}$ | $65.0_{\pm1.3}$ | $15.0_{\pm7.1}$ | $30.0_{\pm0.0}$ | $13.3_{\pm9.4}$ | $36.7_{\pm4.7}$ | $47.2_{\pm0.5}$ | $60.0_{\pm0.9}$ |
| | IPPO ($T=2$) | $84.1_{\pm0.5}$ | $89.8_{\pm0.4}$ | $73.6_{\pm0.6}$ | $79.8_{\pm0.7}$ | $60.0_{\pm4.5}$ | $70.5_{\pm4.0}$ | $17.5_{\pm2.1}$ | $29.4_{\pm2.5}$ | $19.5_{\pm2.3}$ | $36.1_{\pm3.2}$ | $50.9_{\pm2.1}$ | $61.1_{\pm2.3}$ |
| | UMAD ($T=2$) | $90.0_{\pm0.7}$ | $90.7_{\pm0.4}$ | $83.3_{\pm0.3}$ | $82.7_{\pm1.0}$ | $65.8_{\pm6.3}$ | $62.5_{\pm2.5}$ | $20.0_{\pm8.8}$ | $28.9_{\pm1.9}$ | $26.7_{\pm8.8}$ | $35.6_{\pm5.1}$ | $57.2_{\pm3.6}$ | $60.1_{\pm1.0}$ |
| | Zero-Shot ($T=5$) | $87.3_{\pm0.5}$ | $90.5_{\pm0.4}$ | $78.2_{\pm0.6}$ | $84.6_{\pm0.4}$ | $60.0_{\pm1.4}$ | $75.0_{\pm1.2}$ | $28.3_{\pm2.4}$ | $\mathbf{38.3}_{\pm7.1}$ | $20.0_{\pm0.0}$ | $40.0_{\pm4.7}$ | $54.8_{\pm0.5}$ | $65.7_{\pm0.5}$ |
| | IPPO ($T=5$) | $88.3_{\pm0.4}$ | $90.3_{\pm0.3}$ | $81.0_{\pm0.5}$ | $85.1_{\pm0.6}$ | $70.5_{\pm3.8}$ | $\mathbf{82.5}_{\pm3.2}$ | $30.2_{\pm2.2}$ | $36.4_{\pm2.9}$ | $31.1_{\pm2.5}$ | $42.8_{\pm3.1}$ | $60.2_{\pm1.8}$ | $\mathbf{67.4}_{\pm2.0}$ |
| | UMAD ($T=5$) | $\mathbf{91.1}_{\pm0.6}$ | $\mathbf{91.1}_{\pm0.2}$ | $\mathbf{87.0}_{\pm0.7}$ | $\mathbf{86.1}_{\pm0.6}$ | $\mathbf{80.0}_{\pm4.3}$ | $71.7_{\pm3.8}$ | $\mathbf{32.2}_{\pm6.9}$ | $34.4_{\pm5.1}$ | $\mathbf{42.2}_{\pm5.1}$ | $\mathbf{45.6}_{\pm7.7}$ | $\mathbf{66.5}_{\pm2.8}$ | $65.8_{\pm3.1}$ |

## 5.2. Main Results

Our results show that uncertainty-guided training improves heterogeneous debate primarily through asymmetric gains for the weaker agent (Table 1). In the heterogeneous setting, UMAD raises the weaker `Qwen2.5-3B-Instruct` agent from $54.8$ to $66.5$ average accuracy at $T = 5$, outperforming both Zero-Shot MAD and Standard IPPO. The stronger `Qwen3-4B-Instruct-2507` agent remains close to its Zero-Shot performance but is not uniformly improved, indicating that the main benefit of UMAD is not simply increasing standalone accuracy for all agents. Instead, the gains suggest improved utilization of complementary peer information by the weaker model.

In homogeneous settings, the training gains of both IPPO and UMAD are modest and largely saturate, suggesting a ceiling when epistemic diversity is limited. The improvements over zero-shot MAD baselines are small, which is consistent with the view that homogeneous debate mainly acts as self-consistency (Choi et al., 2026). Without sufficiently non-redundant peer information, extensive epistemic overlap constrains progress. Notably, UMAD provides small but stable gains, consistent with the view that homogeneous debate has limited non-redundant epistemic information.

Finally, Table 1 shows that baseline methods tend to saturate or degrade as the number of debate rounds increases, leading to saturation or decline, whereas UMAD remains robust as $T$ increases. The consistent improvements up to $T = 5$, especially in heterogeneous and out-of-distribution tasks, indicate that uncertainty-guided training improves robustness to longer debate horizons. By down-weighting unstable responses and rewarding useful peer influence, agents sustain constructive long-context interaction.

## 5.3. Ablation Analysis

**Effectiveness of Uncertainty Guidance.** Comparing UMAD with Standard IPPO isolates the effect of uncertainty-guided optimization. In heterogeneous debate, UMAD achieves the largest improvement for the weaker agent, especially at $T = 5$, improving its average accuracy from $60.2$ under IPPO to $66.5$. This suggests that uncertainty-guided rewards improve peer-information utilization beyond standard multi-agent RL. Additional ablations in Appendix E.1 further show that NLL-only and Intrinsic-only training underperform Full UMAD on the weaker agent, indicating that aleatoric control and epistemic influence are complementary. We also compare with independently trained Single-GRPO agents; although this stronger baseline improves the stronger agent, UMAD achieves higher post-debate accuracy for the weaker agent, suggesting that debate co-training improves heterogeneous knowledge absorption beyond standalone RL. We observe that standard IPPO agents can converge to lazy consensus with short responses such as *"I agree with the other agent's answer."*, whereas UMAD encourages more constructive peer influence on harder problems.

**Generalization to Longer Debates.** Although UMAD is trained with short two-round debates ($T_{\text{train}} = 2$), it generalizes to $T = 5$ inference-time debates without divergence. For the weaker agent in heterogeneous debate, the average accuracy further increases from $57.2$ at $T = 2$ to $66.5$ at $T = 5$. This indicates that the learned policy is not merely overfitted to a fixed trajectory length. Instead, UMAD learns a reusable propose-refine interaction pattern that remains effective under longer debate horizons.

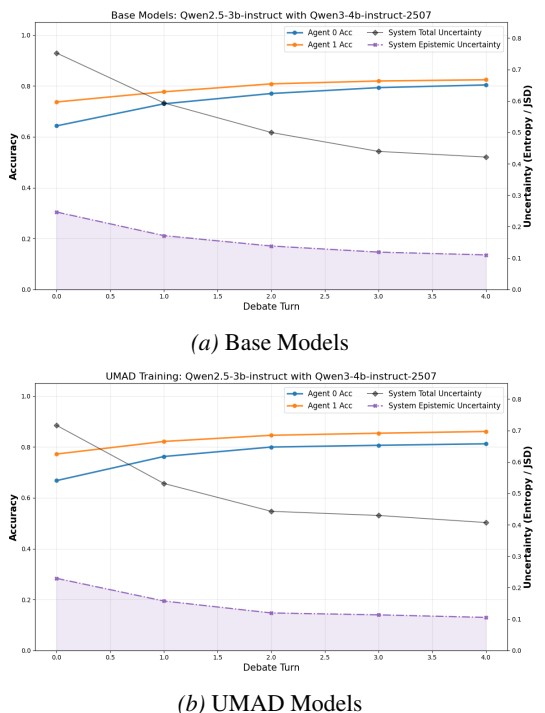

*(a)* Base Models

*(b)* UMAD Models

*Figure 3.* Uncertainty dynamics (gray and purple) vs. accuracy (orange and blue). UMAD training suppresses aleatoric noise and improves both agents' accuracies across debate turns.

**Visualizing Uncertainty Dynamics.** Fig. 3 supports our uncertainty-based interpretation. After UMAD training, aleatoric uncertainty decreases across debate rounds, while the weaker agent's accuracy changes from stagnation to steady improvement. This suggests that the weaker agent benefits from more stable utilization of the stronger peer's reasoning, rather than merely conforming to peer outputs.

## 6. Related Works

**Multi-Agent Debate.** Prior works in MAD (Du et al., 2024; Liang et al., 2024; Chan et al., 2024) show that iterative debates can improve reasoning over single-agent prompting methods such as CoT (Wei et al., 2022) and Self-Consistency (Wang et al., 2023). Existing studies mainly improve MAD through workflow design (Liu et al., 2024; Zhang & Xiong, 2025; Maiga et al., 2025; Yao et al., 2025), communication topology optimization (Li et al., 2024; Sun et al., 2025; Eo et al., 2025), model-combination analysis (Zhang et al., 2025; Smit et al., 2024), or confidence-aware prompting and aggregation (Yang et al., 2024b; Chen et al., 2024; Lin & Hooi, 2025; Yoffe et al., 2025). However, recent findings show that inference-time debate can suffer from sycophancy, long-context drift, and consensus without correctness (Choi et al., 2026; Pan et al., 2025; Yang et al., 2025b; Wynn et al., 2025). Unlike methods that rely on frozen-model prompting or aggregation rules, we analyze answer-level uncertainty during debate and use uncertainty-guided training signals to improve peer information utilization.

**Diversity and Uncertainty in MAD.** Concurrent studies have also examined diversity and uncertainty in MAD. Zhu et al. (2026) attribute MAD failures to low initial diversity and uncalibrated confidence, while Ai et al. (2025); Li et al. (2026b) optimize aggregation or reasoning paths to improve information gains. Other works highlight limitations of heterogeneous teams, including the leveraging gap and integrative compromise (Li et al., 2026a; Pappu et al., 2026), or explain heterogeneity through complementary evidence (Yang et al., 2026). These works mainly analyze inference-time behavior or aggregation protocols. In contrast, our work connects uncertainty decomposition to trainable debate behavior through aleatoric-control and epistemic-influence proxies. This enables the system to actively manage the trade-off between exploring diverse peer perspectives and maintaining individual reasoning stability. Consequently, we transform empirical debate heuristics into a formal, optimization-driven training process.

**Uncertainty in LLMs and Multi-Agent RL.** Uncertainty quantification has been widely studied for LLM trustworthiness (Huang et al., 2025), including token-level confidence, semantic uncertainty (Kuhn et al., 2023; Wang et al., 2025), and Bayesian decompositions into aleatoric and epistemic uncertainty (Liu et al., 2025b; Huang et al., 2024; Geng et al., 2024; Abbasi Yadkori et al., 2024; Jayasekera et al., 2025; Ling et al., 2024). Related work also uses multi-agent perturbations (Amayuelas et al., 2024) or fine-tuning (Feng et al., 2025) to elicit uncertainty signals. Separately, recent MARL methods train LLM-based multi-agent systems through role design, policy optimization, or credit assignment (Park et al., 2025; Liao et al., 2025; Liu et al., 2026; Zhao et al., 2026). Our method combines these directions by using uncertainty diagnostics within MAD to design MARL rewards that stabilize individual reasoning and encourage useful peer influence. More related work is discussed in Appendix A.

## 7. Conclusion

In this work, we address the inherent instability and inefficiency of Multi-Agent Debate, particularly in heterogeneous settings, through a Bayesian uncertainty lens. We identify that performance degradation stems from uncontrolled aleatoric noise and the failure to effectively utilize epistemic disagreement. To resolve this, we propose an uncertainty-guided MARL framework that actively regulates internal reasoning confidence and incentivizes constructive information exchange. Our approach enables robust performance gains and prevents belief collapse in long-turn debates, transforming MAD from a fragile inference-time heuristic into a stable, learnable mechanism for reliable reasoning.

# Acknowledgements

Baoxiang Wang and Dan Qiao are partially supported by the National Natural Science Foundation of China (No. 72394361) and the Shenzhen Science and Technology Program (Nos. JCYJ20250604141218024 and JCYJ20250604141032005). Wenhao Li is supported by the NSFC (No. 62406270) and the STCSM Shanghai Rising-Star Program (No. 24YF2748800). Fengyu Cai is funded by the German Federal Ministry of Education and Research as part of the Software Campus 3.0 project ETRAG (funding code 16IS23067). The authors would like to express their sincere gratitude to Youliang Yuan, Yajiao Liu, Fei Yu, Xiaoyuan Liu, Jiawei Xu, Yue Lin, Han Wang, Ang Li, Zhongxiang Dai, and Zhiwei Shang from the Chinese University of Hong Kong, Shenzhen (CUHKSZ) for their insightful discussions and constant support. We are also grateful to Lihu Chen from Imperial College London for his valuable feedback during the early development of this work. Finally, we thank the anonymous reviewers for their constructive comments and suggestions.

# Impact Statement

This work aims to enhance the reliability of Large Language Models by grounding multi-agent debate in a Bayesian framework. By mitigating confident hallucinations, our method (UMAD) supports the deployment of trusted AI in critical fields like mathematics and science. However, we acknowledge that explicitly optimizing agents for "persuasiveness" carries potential risks; if applied to subjective or sensitive domains, this mechanism could theoretically be misused to generate convincing but manipulative content. Future research should ensure such capabilities remain strictly aligned with factual verification. Additionally, we address the environmental footprint of multi-agent systems by demonstrating that efficient, short-horizon training generalizes effectively to long-context inference.

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

# A. More Related Works

**Multi-Agent Debate, Confidence, and Debate Failures.**   Early MAD studies demonstrate that iterative interaction among multiple LLM agents can improve reasoning by eliciting diverse solution paths and aggregating final answers (Du et al., 2024; Liang et al., 2024; Chan et al., 2024). Subsequent work improves MAD through task-specific workflows (Liu et al., 2024; Zhang & Xiong, 2025; Maiga et al., 2025; Yao et al., 2025), communication topologies (Li et al., 2024; Sun et al., 2025; Eo et al., 2025), and model-combination analysis (Zhang et al., 2025; Smit et al., 2024). Another line of work introduces confidence signals into debate, either by adding peer confidence to prompts (Yang et al., 2024b; Chen et al., 2024) or by using confidence as aggregation weights (Lin & Hooi, 2025; Eo et al., 2025; Yoffe et al., 2025). However, recent studies show that inference-time debate does not always improve individual reasoning. Homogeneous debates can stagnate after removing aggregation (Choi et al., 2026), agents may flip correct answers to incorrect ones (Wynn et al., 2025), and sycophancy or long-context drift can push agents toward superficial agreement (Pan et al., 2025; Yang et al., 2025b). These findings motivate our focus on individual belief evolution rather than final aggregation accuracy.

**Diversity and Uncertainty in MAD.**   Several concurrent studies analyze the role of diversity, confidence, and uncertainty in MAD. Zhu et al. (2026) argue that low initial diversity and uncalibrated confidence are key causes of MAD failures, and propose a textual-confidence-modulated protocol to mitigate sycophancy. To quantify the behavioral biases that disrupt collective reasoning, Choi et al. (2025) formalize debate dynamics as an identity-weighted Bayesian update process and propose response anonymization to enforce equal weighting on agent identities. Similarly, Ai et al. (2025) study higher-order aggregation rules for combining multiple responses, while Li et al. (2026b) dynamically generates diverse reasoning paths to break homogeneous reasoning patterns. Benchmark studies such as Li et al. (2026a) question whether MAD is consistently effective across domains and modalities. Pappu et al. (2026) further show Graves-agent teams may underutilize their strongest members due to a leveraging gap and integrative compromise. Complementarily, Yang et al. (2026) provides an information-theoretic explanation of agent scaling through diversity and complementary evidence. Our work differs from these studies by linking diversity and uncertainty to an explicit epistemic-gain versus aleatoric-cost decomposition, and by using this decomposition to design training-time objectives rather than only inference-time protocols, anonymization rules, or aggregation mechanics.

**Uncertainty Quantification in LLMs.**   Uncertainty quantification is central to assessing LLM reliability (Huang et al., 2025). Token-level probabilities provide a convenient confidence signal but often fail to capture semantic equivalence or reasoning uncertainty. Semantic uncertainty methods address this limitation by clustering semantically equivalent generations and measuring uncertainty over meaning rather than surface forms (Kuhn et al., 2023; Wang et al., 2025). From an aggregation perspective, Choi & Li (2026) leverage semantic consensus across multi-path generations by constructing similarity graphs and applying spectral clustering to achieve evaluator-free Best-of-N selection. Bayesian uncertainty decompositions further distinguish epistemic uncertainty, associated with model knowledge or reducible uncertainty, from aleatoric uncertainty, associated with inherent ambiguity or response variability (Liu et al., 2025b; Huang et al., 2024; Geng et al., 2024). These ideas have been explored in general question answering (Abbasi Yadkori et al., 2024), in-context learning (Jayasekera et al., 2025; Ling et al., 2024), multi-agent perturbation methods (Amayuelas et al., 2024), and uncertainty-aware fine-tuning (Feng et al., 2025). In contrast, we study uncertainty as a dynamic property of the MAD process itself, where cross-agent disagreement and within-agent response instability evolve over debate turns.

**Multi-Agent Reinforcement Learning in LLMs.**   With the development of LLM-based multi-agent systems, recent research has moved from heuristic workflow design toward trainable multi-agent learning paradigms. Early explorations often focused on interaction-driven data generation or task-specific role design. For instance, SPIRAL (Liu et al., 2025a) demonstrated that reasoning capabilities can emerge from self-play in zero-sum games, though its primary focus is action-based tasks rather than mathematical reasoning. Multi-Agent Finetuning (MAFT) (Subramaniam et al., 2025) uses multi-agent interactions to collect diverse reasoning chains for supervised fine-tuning. ReMA (Wan et al., 2026) introduces a hierarchical architecture that alternates between a "Meta-thinking Agent" and a "Reasoning Agent" to decouple strategic oversight from execution. More recent studies formulate multi-agent reasoning as a reinforcement learning problem. MAPoRL (Park et al., 2025) formalizes debate as a MARL process and adopts an IPPO-style (De Witt et al., 2020) training objective to reward persuasive and corrective behaviors. MARFT (Liao et al., 2025) proposes the LaMAS framework, motivated by HAPPO (Kuba et al., 2022), to improve policy updates for LLM-based agents. Other methods focus on credit assignment and group dynamics. MAGRPO (Liu et al., 2026) introduces a centralized team-reward style MARL method based on MAPPO (Yu et al., 2022), while AT-GRPO (Zhao et al., 2026) uses agent- and turn-aware grouping to handle

non-stationarity in multi-turn dialogues. Further research explores consensus-based alignment in MACA (Samanta et al., 2025) and meta-policy deliberation in MPDF (Yang & Thomason, 2026). Systems such as MARTI (Zhang et al., 2026) provide scalable infrastructure for asynchronous multi-agent rollouts. Most of these MARL studies focus on homogeneous settings, role prompting, or general credit assignment, and do not explicitly analyze the uncertainty trade-off that governs debate success. Concurrently, Tang et al. (2026) address debate collapse by proposing a hierarchical uncertainty metric and formulating an uncertainty-driven policy optimization to penalize behavioral volatility and peer conflicts. In contrast, UMAD connects MARL training to an explicit epistemic-gain versus aleatoric-cost decomposition of MAD. Its epistemic influence reward encourages responses that improve peer reasoning, while its aleatoric-uncertainty-aware advantage shaping suppresses unstable generations. This makes the training objective directly aligned with the uncertainty mechanism identified in our analysis.

## B. Theoretical Analysis

### B.1. Proof of Lemma 3.1

**Lemma Statement.** *Assume $p(h \mid x, c \oplus m) = p(h \mid x, c, m)$ and $p(m \mid x, c) > 0$. Then:*

$$\log \frac{p(h = 1 \mid x, c \oplus m)}{p(h = 0 \mid x, c \oplus m)} = \log \frac{p(h = 1 \mid x, c)}{p(h = 0 \mid x, c)} + \log \frac{p(m \mid h = 1, x, c)}{p(m \mid h = 0, x, c)}.$$

*Proof.* Recall Bayes' theorem:

$$p(h \mid x, c, m) = \frac{p(m \mid h, x, c) \cdot p(h \mid x, c)}{p(m \mid x, c)}. \tag{6}$$

Writing this for $h = 1$ and $h = 0$ separately:

$$p(h = 1 \mid x, c, m) = \frac{p(m \mid h = 1, x, c) \cdot p(h = 1 \mid x, c)}{p(m \mid x, c)}, \tag{7}$$

$$p(h = 0 \mid x, c, m) = \frac{p(m \mid h = 0, x, c) \cdot p(h = 0 \mid x, c)}{p(m \mid x, c)}. \tag{8}$$

Dividing the first equation by the second cancels out the marginal likelihood $p(m \mid x, c)$:

$$\frac{p(h = 1 \mid x, c, m)}{p(h = 0 \mid x, c, m)} = \frac{p(h = 1 \mid x, c)}{p(h = 0 \mid x, c)} \cdot \frac{p(m \mid h = 1, x, c)}{p(m \mid h = 0, x, c)}. \tag{9}$$

Taking the natural logarithm on both sides yields the additive log-odds update:

$$\log \frac{p(h = 1 \mid x, c \oplus m)}{p(h = 0 \mid x, c \oplus m)} = \log \frac{p(h = 1 \mid x, c)}{p(h = 0 \mid x, c)} + \log \frac{p(m \mid h = 1, x, c)}{p(m \mid h = 0, x, c)}. \tag{10}$$

This concludes the proof. $\square$

### B.2. Proof of Proposition 3.2

**Proposition Statement.** *Let $\text{TU}(t) := H(\bar{p}_t)$. Then $\text{TU}(t) = \text{Sys-EU}(t) + \text{Sys-Cost}(t)$, where $\text{Sys-EU}(t) = \text{JSD}(p_{1,t}, \ldots, p_{N,t})$ and $\text{Sys-Cost}(t) = \frac{1}{N} \sum_i H(p_{i,t})$.*

*Proof.* This follows directly from the definition of the Generalized Jensen-Shannon Divergence (JSD). For a set of distributions $\{p_1, \ldots, p_N\}$ and weights $\pi_1 = \cdots = \pi_N = \frac{1}{N}$, the JSD is defined as:

$$\text{JSD}(p_1, \ldots, p_N) := H\left(\sum_{i=1}^{N} \pi_i p_i\right) - \sum_{i=1}^{N} \pi_i H(p_i). \tag{11}$$

Substituting our notation $\bar{p}_t = \frac{1}{N} \sum_i p_{i,t}$:

$$\text{Sys-EU}(t) = H(\bar{p}_t) - \frac{1}{N} \sum_{i=1}^{N} H(p_{i,t}). \tag{12}$$

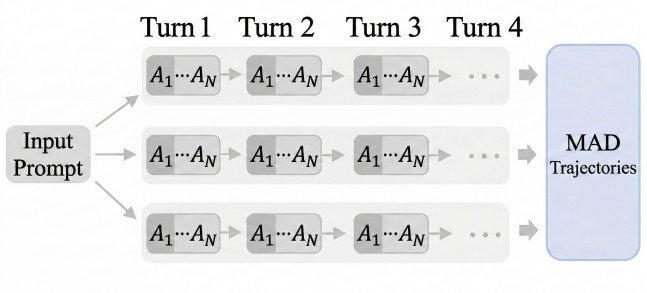

*Figure 4.* Trajectory-level pairwise debate rollouts strategy.

Rearranging the terms:

$$H(\bar{p}_t) = \text{Sys-EU}(t) + \frac{1}{N}\sum_{i=1}^{N} H(p_{i,t}). \tag{13}$$

By definition, $\text{TU}(t) = H(\bar{p}_t)$ and $\text{Sys-Cost}(t) = \frac{1}{N}\sum_i H(p_{i,t})$, which proves the identity. □

### B.3. Proof of Theorem 3.4

*Proof.* By the chain rule of conditional mutual information, the total epistemic information provided by two homogeneous messages can be decomposed as

$$I(\varphi; m_{\text{homo}}, m'_{\text{homo}} \mid x, c) = I(\varphi; m_{\text{homo}} \mid x, c) + I(\varphi; m'_{\text{homo}} \mid x, c, m_{\text{homo}}). \tag{14}$$

Similarly, replacing the second homogeneous message with a heterogeneous message gives

$$I(\varphi; m_{\text{homo}}, m_{\text{hetero}} \mid x, c) = I(\varphi; m_{\text{homo}} \mid x, c) + I(\varphi; m_{\text{hetero}} \mid x, c, m_{\text{homo}}). \tag{15}$$

Subtracting the two identities yields

$$\begin{aligned} &I(\varphi; m_{\text{homo}}, m_{\text{hetero}} \mid x, c) - I(\varphi; m_{\text{homo}}, m'_{\text{homo}} \mid x, c) \\ &= I(\varphi; m_{\text{hetero}} \mid x, c, m_{\text{homo}}) - I(\varphi; m'_{\text{homo}} \mid x, c, m_{\text{homo}}). \end{aligned} \tag{16}$$

By Assumption 3.3,

$$I(\varphi; m'_{\text{homo}} \mid x, c, m_{\text{homo}}) \leq \epsilon_{\text{homo}},$$

and by the heterogeneous complementarity condition,

$$I(\varphi; m_{\text{hetero}} \mid x, c, m_{\text{homo}}) \geq \epsilon_{\text{homo}} + \Delta.$$

Therefore,

$$I(\varphi; m_{\text{homo}}, m_{\text{hetero}} \mid x, c) - I(\varphi; m_{\text{homo}}, m'_{\text{homo}} \mid x, c) \geq \Delta,$$

which proves the claim. □

## C. Algorithms and Pseudocode

### C.1. Uncertainty Quantification of Multi Agent Debate

In the context of multi-agent debate, standard step-level branching leads to a combinatorial explosion of interaction paths—a computational bottleneck highlighted by (Feng et al., 2026). To mitigate this, we adopt a 'Trajectory-level Pairwise Debate Rollouts' strategy that maintains $K$ parallel debate histories as shown in Fig. 4. Specifically, at the initial turn ($t = 0$), each participating agent generates $K = 16$ independent rollouts. These responses are then synchronized by index to form $K$ distinct conversation threads (i.e., the $k$-th response of Agent A is paired with the $k$-th response of Agent B). For all subsequent turns ($t > 0$), agents generate a single rollout $K = 1$ for each of these established trajectories. This

---

**Algorithm 1** MAD Uncertainty Quantification

---

1: **Input:** Problems Dataset $\mathcal{D}$, Agents $\{\pi_{\theta_i}\}_{i=1}^N$, Max Turns $T$, Sample Count $K = 16$
2: **for** each problem $x$ in $\mathcal{D}$ **do**
3:     Initialize contexts $c_{i,0} \leftarrow \emptyset$ for all $i$
4:     **for** turn $t = 1$ to $T$ **do**
5:         *// Step 1: Estimate Uncertainty via Sampling*
6:         **for** agent $i = 1$ to $N$ **do**
7:             Generate $K$ sample paths: $\{y_{i,t}^{(k)}\}_{k=1}^K \sim \pi_{\theta_i}(\cdot \mid x, c_{i,t-1})$
8:             Extract answers and estimate distribution $p_{i,t}(y)$ based on count frequencies
9:             Calculate Agent Aleatoric Entropy: $U_{i,t}^{ale} = H(p_{i,t})$
10:         **end for**
11:         *// Step 2: Calculate System Uncertainty Metrics (Eq. 3)*
12:         Calculate Mixture Distribution: $p_{\text{sys},t}(y) = \frac{1}{N} \sum_{i=1}^N p_{i,t}(y)$
13:         **System TU:** $\text{TU}_t = H(p_{\text{sys},t})$
14:         **System AU:** $\text{Sys-AU}_t = \frac{1}{N} \sum_{i=1}^N U_{i,t}^{ale}$
15:         **System EU (JSD):** $\text{Sys-EU}_t = \text{TU}_t - \text{Sys-AU}_t$
16:         *// Step 3: Proceed to Next Round (Single Trajectory)*
17:         **for** agent $i = 1$ to $N$ **do**
18:             Sample a single response $\hat{y}_{i,t} \sim \pi_{\theta_i}(\cdot \mid x, c_{i,t-1})$ *// or select greedily*
19:         **end for**
20:         Update Context: $c_{i,t} \leftarrow \Phi_i(\{\hat{y}_{j,t}\}_{j=1}^N)$
21:     **end for**
22: **end for**

---

mechanism ensures that each of the $K$ trajectories evolves linearly with consistent context, enabling efficient GRPO without the exponential overhead of recursive branching.

Then we utilize these $K$ parallel debate rollouts to approximate the implicit belief distribution by aggregating the frequency of final answers. To ensure robustness, we extract the answer key from each rollout and merge semantically equivalent terms. Crucially, to enable valid divergence calculations across agents who may sample disjoint answer sets, we define the common support set $\mathcal{Y}_{union}$ as the union of all unique answers generated by the entire team at turn $t$. Each agent $i$'s empirical distribution $\hat{p}i, t(y)$ is then computed over this global support $\mathcal{Y}_{union}$, assigning zero probability to unobserved outcomes. Based on these aligned distributions, we quantify the uncertainty using the team's mean distribution $\bar{p}_t(y) = \frac{1}{N} \sum_{i=1}^N \hat{p}i, t(y)$. Specifically, the **Total Uncertainty** is defined as the entropy of the team mean $H(\bar{p}_t)$. This is further decomposed into **Epistemic Uncertainty**, and **Aleatoric Uncertainty**, measured by the Jensen-Shannon Divergence among agents and the average entropy of individual distributions. The pseudo-code is demonstrated in Table 1.

## C.2. UMAD Training Algorithm

Algorithm 2 presents the Uncertainty-Guided Multi-Agent Debate (UMAD) training framework, which extends GRPO to optimize collaborative reasoning. The procedure involves sampling a group of joint debate trajectories to estimate advantage baselines, incorporating two key regulatory mechanisms: an Epistemic Influence Intrinsic Reward that credits agents for positively shaping peer performance in subsequent turns, and an Aleatoric Uncertainty-Aware Advantage that modulates gradient updates based on token-level confidence. By integrating these components, the algorithm optimizes the policy to minimize internal reasoning noise while maximizing the effective information gain provided to the multi-agent system.

# D. Experiment Details

## D.1. Datasets

**MATH** (Hendrycks et al., 2021): The MATH dataset contains 12,500 challenging competition mathematics problems including algebra, geometry, combinatorics, and number theory. Each problem is provided with a full step-by-step solution with multi-step reasoning process.

---

**Algorithm 2** Uncertainty-Guided Multi-Agent Debate Training (UMAD)

---

1: **Input:** Dataset $\mathcal{D}$, Policy $\pi_\theta$, Ref Model $\pi_{ref}$, Learning Rate $\alpha$, Group Size $G$, Debate Turns $T_{train}$
2: **Hyperparams:** KL coeff $\beta$, Epistemic Reward weight $\eta$
3: **while** not converged **do**
4:  Sample batch of questions $x \sim \mathcal{D}$
5:  **for** each $x$ **do**
6:   *// Collect Trajectories via Debate Rollout*
7:   **for** group rollout $g = 1$ to $G$ **do**
8:    Initialize $c_{i,0} = \emptyset$
9:    **for** turn $t = 1$ to $T_{train}$ **do**
10:     **for** agent $i = 1$ to $N$ **do**
11:      Sample response $y_{i,t} \sim \pi_\theta(\cdot \mid x, c_{i,t-1})$
12:      Compute token-level log-probs for $\hat{\mathcal{U}}_i$
13:      Verify correctness $R_{corr}(y_{i,t}, y^*) \in \{0, 1\}$
14:     **end for**
15:     Update Context $c_t$ with peer responses
16:    **end for**
17:   **end for**
18:   *// Compute Rewards and Advantages*
19:   **for** each response $(y_{i,t})$ in batch **do**
20:    *1. Calculate Epistemic Intrinsic Reward*
21:    Calculate Mean Peer Improvement $\Delta R_{\text{peer}}$ based on next turn correctness
22:    Total Reward $R_{total} = R_{corr} + \eta \cdot \Delta R_{\text{peer}}$ (if $t < T_{train}$)
23:    *2. Compute Group Advantage*
24:    $A_{i,t} = \frac{R_{total} - \text{mean}(R_{group})}{\text{std}(R_{group}) + \epsilon}$
25:    *3. Apply Aleatoric Uncertainty Weighting*
26:    $A_{i,t}^{au} = W(\hat{\mathcal{U}}_i) \cdot A_{i,t}$
27:   **end for**
28:   *// GRPO Update*
29:   $\mathcal{L} = \mathbb{E}\left[\min(rA^{au}, \text{clip}(r)A^{au}) - \beta D_{KL}(\pi_\theta \| \pi_{ref})\right]$
30:   Update $\theta \leftarrow \theta - \alpha \nabla_\theta \mathcal{L}$
31:  **end for**
32: **end while**

---

**GSM8K** (Cobbe et al., 2021): GSM8K contains 8,500 grade-school-level word problems focusing on arithmetic and logical reasoning.

**AMC-23** (Knovel Engineering, 2025): This dataset contains 40 expert mathematical problems drawn from the 2023 American Mathematics Competitions, including spanning algebra, combinatorics, geometry, and number theory.

**AIME24 (25)** (Art of Problem Solving, 2024; 2025): Each dataset contains 30 competition-level problems from the American Invitational Mathematics Examination I and II tests.

### D.2. LLM Configurations and Hyperparameters

We implement our Uncertainty-Guided Multi-Agent Debate (UMAD) training based on the Group Relative Policy Optimization (GRPO) framework (Shao et al., 2024). The training is conducted using PyTorch with the veRL (?) on $8\times$ NVIDIA H800 GPUs. The detailed hyperparameters for generation, optimization, and the GRPO algorithm are listed in Table 2.

**Reward Configuration.** For the correctness reward, we assign $+1$ for the correct final answer and $0$ otherwise. The Epistemic Influence intrinsic reward is controlled by $r_{i,t}^{eu} = 0.25$ to balance the trade-off between self-correctness and peer-persuasion. The exact match is determined using the strict equivalence reward script provided by the MATH dataset benchmark (Hendrycks et al., 2021).

**Prompt Formatting.** We adopt the standard chat template of math reasoning problems. The debate context from previous

*Table 2.* Detailed Hyperparameters for UMAD Training and Inference.

| Hyperparameter | Value |
|---|---|
| *Generation Configuration* | |
| Training Temperature | 0.8 |
| Inference Temperature | 0.6 |
| Top-p | 0.95 |
| Max Response Length | 2048 |
| Max Prompt Length | 5120 |
| *Optimizer Configuration* | |
| Optimizer | Adam |
| Learning Rate | $1 \times 10^{-6}$ |
| Weight Decay | 0.01 |
| Gradient Clipping Norm | 1.0 |
| Clip Ratio | 0.2 |
| *GRPO Training Configuration* | |
| Training Epochs | 2 |
| Train Batch Size | 256 |
| Training Debate Rounds ($T_{train}$) | 2 |
| Evaluation Debate Rounds ($T_{test}$) | 5 |
| Group Size ($G$) | 5 |
| KL Coefficient ($\beta$) | 0.001 |
| Epistemic Influence Intrinsic Rewards ($r_{i,t}^{eu}$) | 0.25 |
| Aleatoric Uncertainty Advantage Ratio ($\alpha_{au}$) | 0.25 |

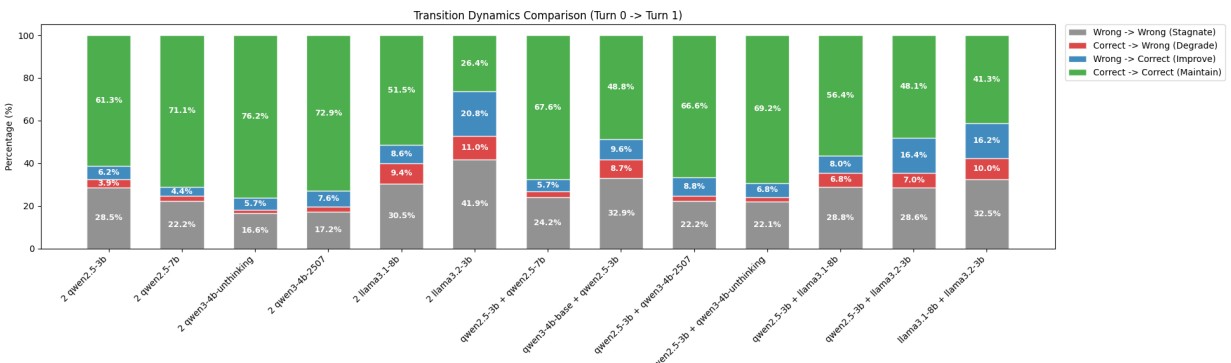

*Figure 5.* Homogeneous and Heterogeneous First Turn Responses Flipping Ratio.

rounds is appended to the user message history to ensure the model attends to the peer's reasoning paths. The detailed prompt templates are demonstrated in Section F.

### D.3. Debate Configurations

In this paper, we set each agent's response length as 2,048 and prompt length as 5,120. Based on grid search of MAD with different length settings, we found that 2,048 is enough for handling most questions with debate contexts, the accuracy of general setting as 8k or 32k is marginally above the 2k setting. Considering the computational cost, we set 2k as default settings for both training and evaluation periods. For AIME problems, we extend the response length to 4k and prompt length as 9k to avoid truncated answers.

### D.4. Dynamics of Answer Flipping

To study how agents navigate increased uncertainty, we analyze answer changes (flipping) from the initial response to the first debate round. We partition outcomes into four transitions: *Correct-to-Correct (C2C)*, *Correct-to-Wrong (C2W)*, *Wrong-to-Correct (W2C)*, and *Wrong-to-Wrong (W2W)*. The *Flip Ratio* is defined as the proportion of responses that change

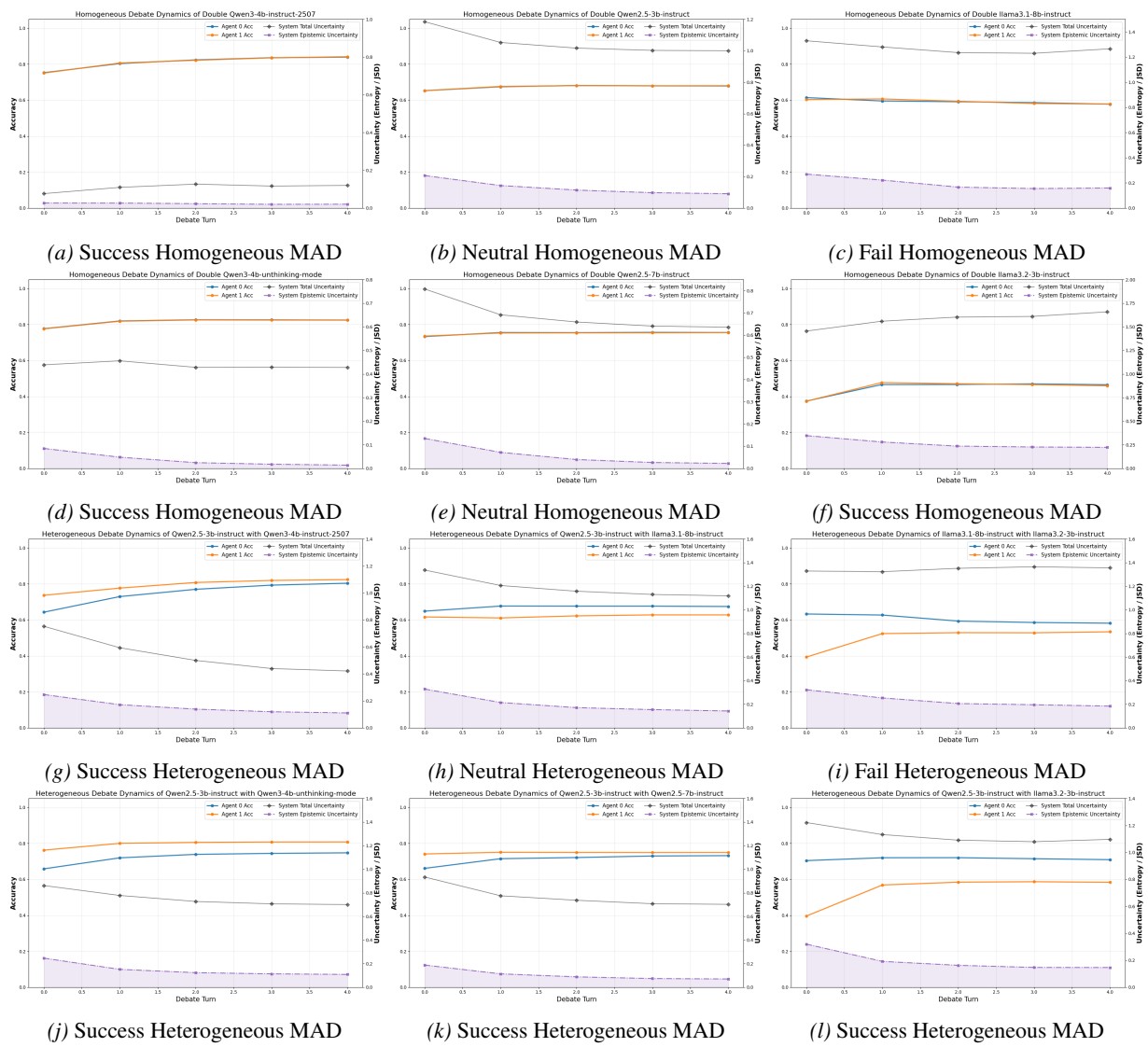

*Figure 6.* Overview of uncertainty decomposition in MAD. The purple dashed area represents the system epistemic uncertainty and the gray line represents the total uncertainty. The blue and orange lines represent the accuracy on the test datasets. **Top two rows:** Homogeneous multi-agent debate. **Bottom two rows:** Heterogeneous multi-agent debate.

their binary correctness state between the two rounds (i.e., $C \rightarrow W$ and $W \rightarrow C$). This statistic is widely used in prior debate analyses to quantify stability and susceptibility to peer influence; our definition aligns with those conventions to ensure direct comparability.

Fig. 5 demonstrates first-round flip ratios for homogeneous and heterogeneous pairs. First, flip ratios are lower for stronger models and higher for weaker models, mirroring the stability differences reported in prior debate studies. Second, heterogeneous pairs exhibit intermediate flip ratios relative to their homogeneous baselines. These observations align with our uncertainty framework: higher flip ratios coincide with higher predictive uncertainty, while lower flip ratios reflect more concentrated beliefs.

### D.5. System-level Uncertainty Decomposition Results

To validate our framework, we conducted experiments across seven models from the Qwen2.5, Qwen3, and Llama families, forming diverse homogeneous and heterogeneous configurations. All debates were conducted on the MATH dataset with $N = 2$ agents for $T = 5$ rounds. Crucially, to bridge the gap between theoretical Bayesian beliefs and empirical

*Table 3.* Three-seed mean and standard deviation for heterogeneous MAD evaluation. We report `pass@1` individual agent accuracy across $T = 1, 2, 5$. Best values are highlighted in bold.

| Method | Round | GSM8K | | MATH500 | | AMC23 | | AIME24 | | AIME25 | | Average | |
|---|---|---|---|---|---|---|---|---|---|---|---|---|---|
| | | A0 | A1 | A0 | A1 | A0 | A1 | A0 | A1 | A0 | A1 | A0 | A1 |
| Full UMAD | $T = 1$ | 83.2±0.9 | 89.5±0.7 | 63.0±0.7 | 79.6±0.9 | 38.3±5.2 | 55.8±3.8 | 3.3±3.3 | 17.8±3.8 | 0.0±0.0 | 27.8±8.4 | 37.6±0.6 | 54.1±2.1 |
| Full UMAD | $T = 2$ | 90.0±0.7 | 90.7±0.4 | 83.3±0.3 | 82.7±1.0 | 65.8±6.3 | 62.5±2.5 | 20.0±8.8 | 28.9±1.9 | 26.7±8.8 | 35.6±5.1 | 57.2±3.6 | 60.1±1.0 |
| Full UMAD | $T = 5$ | **91.1±0.6** | 91.1±0.2 | **87.0±0.7** | 86.1±0.6 | **80.0±4.3** | 71.7±3.8 | **32.2±6.9** | 34.4±5.1 | **42.2±5.1** | **45.6±7.7** | **66.5±2.8** | 65.8±3.1 |
| Intrinsic-only | $T = 1$ | 83.4±0.1 | 89.9±0.2 | 65.3±1.2 | 76.3±1.3 | 39.2±5.2 | 59.2±2.9 | 1.1±1.9 | 18.9±1.9 | 3.3±0.0 | 21.1±1.9 | 38.5±0.8 | 53.1±0.4 |
| Intrinsic-only | $T = 2$ | 86.9±0.6 | 91.5±0.5 | 75.8±1.1 | 83.9±0.4 | 57.5±7.5 | 72.5±2.5 | 14.4±1.9 | 32.2±3.8 | 10.0±3.3 | 35.6±5.1 | 48.9±1.7 | 63.1±2.2 |
| Intrinsic-only | $T = 5$ | 88.1±0.5 | 91.7±0.2 | 78.5±1.4 | 85.4±0.9 | 63.3±9.5 | 83.3±5.2 | 22.2±6.9 | 35.6±1.9 | 15.6±6.9 | 37.8±3.8 | 53.5±4.2 | 66.8±1.5 |
| NLL-only | $T = 1$ | 82.8±1.1 | 91.0±0.4 | 62.1±0.8 | 78.3±0.5 | 38.3±8.8 | 61.7±5.2 | 3.3±3.3 | 15.6±3.8 | 0.0±0.0 | 32.2±1.9 | 37.3±1.8 | 55.7±1.0 |
| NLL-only | $T = 2$ | 87.1±0.9 | 91.0±0.7 | 73.2±0.0 | 81.9±1.2 | 54.2±3.8 | 64.2±6.3 | 11.1±7.7 | 32.2±5.1 | 11.1±1.9 | 38.9±10.2 | 47.3±2.0 | 61.6±1.6 |
| NLL-only | $T = 5$ | 89.4±1.1 | 91.4±0.3 | 79.6±0.5 | 85.4±0.2 | 65.8±1.4 | 68.3±2.9 | 24.4±11.7 | **36.7±6.7** | 20.0±8.8 | 42.2±5.1 | 55.9±1.2 | 64.8±2.2 |
| Single-GRPO Pair | $T = 1$ | 84.7±0.4 | 90.1±0.3 | 63.9±0.1 | 83.1±0.1 | 38.3±7.2 | 66.7±1.4 | 3.3±3.3 | 23.3±3.3 | 1.1±1.9 | 33.3±3.3 | 38.3±2.3 | 59.3±0.8 |
| Single-GRPO Pair | $T = 2$ | 88.1±0.4 | 92.1±0.5 | 74.2±2.5 | 86.9±0.7 | 56.7±2.9 | 82.5±4.3 | 14.4±6.9 | 27.8±5.1 | 15.6±1.9 | 37.8±1.9 | 49.8±1.0 | 65.4±0.9 |
| Single-GRPO Pair | $T = 5$ | 90.4±0.8 | **92.4±0.3** | 83.4±1.0 | **89.5±1.0** | 75.8±3.8 | **88.3±1.4** | 22.2±6.9 | 30.0±3.3 | 28.9±1.9 | 38.9±1.9 | 60.2±2.2 | **67.8±0.5** |

measurement, we set the sampling temperature to 1.0 and generated $K = 16$ sample paths for each agent at every turn. This allows us to explicitly compute the System Total Uncertainty and decompose it into Epistemic Uncertainty (JSD, representing inter-agent conflict) and Aleatoric Uncertainty (Mean Entropy, representing intra-agent instability).

**Analysis of Debate Trajectories.** As illustrated in Fig. 6, our empirical findings reveal that the outcome of a debate is governed by a delicate trade-off between epistemic resolution and aleatoric control:

- **Universal Consensus (Epistemic Decay):** A universal trend across all Successful, Neutral, and Failure trajectories is the monotonic decrease of Epistemic Uncertainty (EU). This suggests that MAD fundamentally functions as a consensus-seeking process regardless of ground-truth correctness, continuously pruning conflicting hypotheses.

- **The "Aleatoric Barrier" to Success:** The divergence in final accuracy is primarily dictated by Aleatoric Uncertainty (AU).
  - *Success Mode:* Agents maintain low or decreasing AU, effectively anchoring valid reasoning against noise.
  - *Failure Mode:* Often observed in weaker models (e.g., Llama series), AU escalates uncontrollably. This indicates that the "contextual cost"—such as hallucination reinforcement—overwhelms the benefits of information exchange.
  - *Neutral Mode:* Represents an equilibrium where epistemic gains are exactly offset by high aleatoric noise, leading to stagnant accuracy.

- **The Heterogeneous Advantage:** Heterogeneous pairs (e.g., Qwen-3B vs. Qwen-4B) exhibit significantly higher initial EU than homogeneous ones. This elevated "cognitive gap" offers greater potential for improvement. Optimal performance is achieved when this high epistemic potential is realized under the stability constraints provided by the stronger model, maximizing effective information gain.

## E. Supplementary Ablation Experiments

### E.1. AU-only and EU-only

We provide additional ablation results in Table 3 to isolate the contribution of each uncertainty component in heterogeneous MAD. All results are reported as three-seed evaluation means and standard deviations using the same heterogeneous setting as the main experiments. Here, *Full UMAD* uses both the epistemic/intrinsic term and the NLL-based aleatoric uncertainty proxy. *Intrinsic-only* removes the AU proxy and keeps only the epistemic/intrinsic component, while *NLL-only* keeps only the NLL-based AU proxy. We also include *Single-GRPO Pair* as a non-decomposed pairwise GRPO baseline. The results show that the two uncertainty terms play complementary roles. Full UMAD achieves the best average performance for the weaker agent A0 at $T = 5$ (66.5±2.8), outperforming Intrinsic-only (53.5±4.2), NLL-only (55.9±1.2), and Single-GRPO Pair (60.2±2.2). This advantage is consistent across GSM8K, MATH500, AMC23, AIME24, and AIME25, suggesting that improving the weaker model in heterogeneous debate requires both informative peer signals and control of noisy contextual influence. Intrinsic-only and NLL-only each capture part of the desired behavior but are insufficient alone. Intrinsic-only performs competitively for the stronger agent A1, reaching 66.8±1.5 average at $T = 5$, but substantially underperforms Full UMAD on A0. This indicates that encouraging epistemic interaction without controlling aleatoric cost may help the

stronger model remain accurate, but does not reliably teach the weaker model to use peer information. Conversely, NLL-only provides a more stable AU-oriented objective and obtains the best A1 result on AIME24, but it lacks the explicit epistemic incentive needed to maximize useful information transfer to A0. Single-GRPO Pair is strong for A1, achieving the best A1 average at $T = 5$ (67.8±0.5), but it remains behind Full UMAD on A0. This pattern is consistent with the main results: standard pairwise optimization can preserve or improve the stronger agent, whereas uncertainty decomposition is more effective for lifting the weaker agent in asymmetric debate. Overall, the ablation supports the design of UMAD as a balanced objective: the epistemic term encourages useful disagreement and information exchange, while the AU proxy suppresses unstable or noisy debate dynamics.

## F. Example Prompts

In this section, we provide the prompts used in the multi-agent debate process, which are uniformly used for all training and evaluation datasets in math reasoning problems.

### F.1. Agent Prompt for Initial Turn 0

In the first round, each agent is provided with the question texts and the standard step-by-step reasoning instructions. The final results are concluded with the format string `\boxed{}`.

> **system** (Could be changed according to specific models' system instructions.)
> You are Qwen, created by Alibaba Cloud. You are a helpful assistant.
> **user**
> `[Question Texts]`
> Let's think step by step and output the final answer within `\boxed{}`.

### F.2. Agent Prompt for Debate Turns

In the following turns, each agent receives the responses of all agents in the last turn as additional reference solutions. All agents' solutions are provided anonymously without specific agent IDs or meaningful roles. This yields a standardized debate setting that avoids role-specific or privileged information and keeps the process grounded in the original task and reference solutions. Instruction prompts encourage agents to refine the reference solutions without imposing a fixed reflection style.

> **system** (Could be changed according to specific models' system instructions.)
> You are Qwen, created by Alibaba Cloud. You are a helpful assistant.
> **user**
> Given the following problem: `[Question Texts]`.
> We have two answers:
> **agent 0** response is: `[Agent 0 reasoning and answer]`.
>
> **agent 1** response is: `[Agent 1 reasoning and answer]`.
>
> Please carefully review these answers and recognize which one is right. If one or all of them are right, please summarize the reasoning process of the right one and give the final answer. If both of them are wrong, please correct their mistakes and provide a novel and complete solution to the problem and give the final answer. Let's think step by step and output the final answer within `\boxed{}`.

## G. Broader Impact

This work advances the understanding of multi-agent interactions in Large Language Models (LLMs) by providing an operational Bayesian framework for uncertainty quantification. By shifting the focus from superficial prompt engineering to the fundamental regulation of internal belief dynamics, our Uncertainty-Guided MARL (UMAD) approach paves the way for more reliable and self-correcting reasoning systems. This has broader implications for deploying LLMs in high-stakes domains (e.g., scientific research, legal analysis) where hallucinations are notoriously risky and "wisdom of crowds" mechanisms are essential for verification.

## H. Computational Resources

All experiments were conducted on a high-performance computing (HPC) cluster. The primary hardware configuration consisted of a single node equipped with $8\times$ NVIDIA H800 (80GB) GPUs. The total computational expenditure is summarized as follows:

- **Inference & Evaluation:** Multi-agent debate evaluation, incorporating aleatoric uncertainty estimation with a sampling scale of $K = 16$, required approximately 40 GPU-hours.

- **UMAD Training:** The Group Relative Policy Optimization (GRPO) training phase for models ranging from 3B to 7B parameters, with a rollout group size of $G = 16$, consumed between 200 and 300 GPU-hours, depending on the specific model scale and context length.

## I. Declarations on Generative AI

In the preparation of this manuscript, generative AI tools were utilized solely for language polishing, including improvements to grammar, clarity, and overall stylistic presentation. All models employed in our experiments are strictly compliant with their respective open-source licenses, and all training and evaluation data were sourced from publicly available datasets, ensuring reproducibility and adherence to ethical data usage standards.

## J. Limitations and Future Works

Our UMAD framework relies on multi-round interactions among multiple agents. Although we show that training on short debates ($T_{\text{train}} = 2$) can generalize to longer inference-time debates ($T_{\text{test}} = 5$), the inference cost still scales approximately linearly with the number of agents and debate rounds, i.e., $N \cdot T$. Future work could explore adaptive stopping, selective communication, or belief-aware routing to improve the accuracy-efficiency trade-off.

Our empirical evaluation focuses on mathematical reasoning, where final answers are verifiable and answer-level uncertainty can be measured reliably. Extending the framework to open-ended tasks, code generation, or scientific reasoning requires more robust answer equivalence checking and uncertainty estimation beyond exact-match verification.

Finally, our uncertainty measures are operational proxies rather than exact Bayesian quantities. We use ensemble answer distributions to approximate system-level predictive uncertainty and token-level negative log-likelihood (NLL) as a practical proxy for response instability. While computationally efficient, token-level NLL is not always aligned with semantic uncertainty, especially for poorly calibrated models or semantically equivalent solutions with different surface forms. Future work could incorporate semantic clustering of generated paths, calibrated confidence estimation, or variational inference to obtain more faithful uncertainty estimates.

