# OpenReview forum: "Epistemic Gain, Aleatoric Cost: Uncertainty Decomposition in Multi-Agent Debate for Math Reasoning"
_ICML.cc/2026/Conference — ICML 2026 regular_

### Official Review · Reviewer_Bq1v · 2026-03-11

**Soundness:** 2
**Presentation:** 2
**Significance:** 3
**Originality:** 3
**Overall Recommendation:** 4
**Confidence:** 3

**Summary:**

This paper investigates the effectiveness mechanism of Multi-Agent Debate (MAD) in mathematical reasoning. By removing the final aggregation step and focusing on individual belief evolution, the authors propose a Bayesian uncertainty decomposition framework that separates system predictive uncertainty into epistemic uncertainty (cross-agent disagreement reducible through information exchange) and aleatoric uncertainty (intra-agent generation noise). Accordingly, they argue that successful MAD depends on high initial epistemic potential and controlled aleatoric cost. Building on this, the authors introduce UMAD, which optimizes multi-agent collaboration through uncertainty-aware advantage and epistemic influence intrinsic rewards. Experiments demonstrate significant improvements for weaker models in heterogeneous pairings, with short-horizon training generalizing to longer debate rounds.

**Compliance With Llm Reviewing Policy:**

Affirmed.

**Final Justification:**

see `Rebuttal Acknowledgement`

**Key Questions For Authors:**

1. Please provide more direct empirical validation demonstrating that these proxies align with the claimed concepts, rather than being artifacts of decoding temperature or answer diversity.

2. Can the current results rule out that "gains primarily stem from additional reward shaping / cooperative RL regularization"? Please provide complete ablations: AU-aware only, epistemic reward only, alternative confidence proxies, and alternative uncertainty metrics.

3. Is Table 1 based on multi-seed training/inference? Please report standard deviations or confidence intervals for the main results.

4. Theorem 3.3's assumption nearly contains the conclusion. Would you be willing to weaken the theoretical claim and explicitly frame it as a "conditional explanatory framework"?

5. Does the method remain valid with more than 2 agents, with aggregation/judge, or on open-ended tasks beyond math?

**Limitations:**

Yes

**Strengths And Weaknesses:**

## Strengths

1. Rather than following the conventional path of designing more complex workflows or aggregation mechanisms, the paper directly investigates the intrinsic mechanisms of MAD, focusing on phenomena such as accuracy improvements accompanied by rising entropy and heterogeneous combinations outperforming homogeneous ones. This aligns with recent discussions on debate failures, sycophancy, and correct-to-wrong flipping.

2. This setup avoids mistaking majority voting gains for debate benefits, enabling a more direct test of whether information exchange genuinely improves individual reasoning.

3. Proposition 3.2 decomposes total entropy into average within-group entropy and JSD. While not theoretically novel, it serves as a natural and computable diagnostic tool for MAD. Empirically, the universal decline of EU and the decisive role of AU control in determining success or failure constitute insightful observations.

4. The argument for heterogeneous debate superiority is well-directed. The authors frame "heterogeneity brings complementarity" as higher epistemic gain, providing a unified language for explaining empirical patterns.

5. The methodological design is relatively complete. UMAD feeds uncertainty diagnosis back into training objectives, balancing confidence calibration with effective influence on others, making the work more complete than pure analysis.

## Weaknesses

Weaknesses

1. Theoretical support is weak. Lemma 3.1 is essentially the log-odds form of Bayes' rule; Proposition 3.2 is a definitional expansion of generalized JSD; Theorem 3.3 derives greater heterogeneous gains under an assumption that nearly contains the conclusion. These help organize the narrative but do not genuinely explain MAD phenomena or derive testable predictions.

2. Key concepts are insufficiently rigorously operationalized. The system treats inter-agent distribution differences as epistemic uncertainty and intra-agent answer entropy as aleatoric uncertainty, which does not naturally hold in LLM math reasoning: diverse answers may stem from phrasing variations, sampling temperature, or reasoning path diversity; the distribution with temperature 1.0 and K=16 samples is highly dependent on decoding choices; the inconsistency between NLL and semantic uncertainty, though acknowledged, lacks stronger validation.

3. A gap exists between reward design and theoretical claims. Aleatoric control uses token mean NLL to weight advantage, which is closer to confidence shaping; epistemic influence reward, based on improving the other's correctness in the next round, is closer to cooperative credit assignment than direct estimation of mutual information.

4. The experimental design is insufficient to rule out simpler explanations. UMAD gains may stem from dense reward/shaping, cooperative training itself, or regularization of output features, rather than uncertainty decomposition correctly capturing the essence of MAD. Key ablations are missing: retaining only AU-aware weighting or peer-influence reward, replacing confidence proxies or uncertainty metrics, etc., leaving mechanism attribution weak.

5. Statistical robustness is inadequate. Table 1 reports only point estimates without standard deviations, confidence intervals, or multiple random seeds. Given the variance in RLVR and MAD, the robustness of 1-3% improvements is difficult to assess, especially on small-data benchmarks like AMC/AIME.

6. Conceptual levels are conflated. EU/AU are first defined at the latent belief level, then proxy-decomposed at the answer-level distributions, with an unclear bridge between them. The conceptual, proxy, and implementation layers should be more clearly distinguished.

7. Differentiation from related work is not sharp enough. The distinctions from "confidence-as-textual-signal," existing multi-agent uncertainty estimation, and "debate as test-time scaling" literature are not sufficiently clarified.

8. Originality lies mainly in combination and framing. Elements such as JSD decomposition, Bayesian narrative, NLL-based confidence shaping, and multi-agent RL are not novel; the contribution lies in integrating them into a unified perspective for explaining MAD dynamics, with empirical insights being more original than the theorems.

---

> ### Author Rebuttal · Authors · 2026-03-31
>
> We sincerely thank Reviewer Bq1v for the incisive critique and for recognizing our effort to investigate the intrinsic mechanisms of MAD rather than simply adding complex aggregation layers. We highly value your feedback regarding our conceptual boundaries and experimental rigor. Here are our responses:
>
> ### 1. Theoretical Support and Theorem 3.3 (W1 & Q4)
> While acknowledging your concerns, these theorems provide a *computable diagnostic framework* for MAD. For instance, Lemma 3.1 formally translates empirical Correct-to-Wrong (C2W) flipping into a trackable log Bayes factor accumulation process. For **Theorem 3.3**, we agree the assumption nearly contains the conclusion. We will explicitly weaken this claim to a **"conditional explanatory framework"** explaining *why* heterogeneity yields superior results when the novelty condition holds, rather than framing it as an unconditional mathematical prediction.
>
> ### 2. Conceptual Levels and Operationalization (W2, W6 & Q1)
>
> Based on your advice, we will explicitly delineate our framework into three layers:
> 1.  **Conceptual Layer:** EU and AU defined at the latent belief level ($\varphi$).
> 2.  **Proxy Layer:** Sys-EU and Sys-AU defined at the answer distribution level (Proposition 3.2), approximating Bayesian model averaging.
> 3.  **Implementation Layer:** Using NLL as an individual AU proxy and peer-correctness-improvement as an epistemic gain proxy.
>
> We fully acknowledge your valid point that this operationalization is highly dependent on decoding choices and risks conflating true epistemic uncertainty with sampling diversity. T=1.0 is intended to expose the predictive distribution, as lower temperatures artificially mask uncertainty, while K=16 is a computational compromise. In the revision, we will explicitly discuss these decoding limitations and clarify that our metrics are practical approximations rather than perfect theoretical mappings.
>
> ### 3. Gap Between Reward Design and Theoretical Claims (W3)
>
> We acknowledge the gap between theoretical EU/AU and practical RL rewards. Since directly estimating mutual information $I(\varphi; y)$ during MARL is computationally intractable, we use directionally aligned proxies:
> * **AU-aware advantage:** Amplifying rewards for high-confidence correct answers and penalizing high-confidence errors directly suppresses intra-agent generation noise (AU).
> * **Epistemic influence reward:** Improving a peer's correctness in the next round is the most direct observable proxy for "epistemic gain."
> We will clarify this approximation gap in the methodology, noting that it functions closer to cooperative credit assignment in practice.
>
> ### 4. Experimental Ablations and Simpler Explanations (W4 & Q2)
>
> To isolate our mechanisms from cooperative RL benefits, note that our **Standard IPPO baseline** (Table 1) serves as a partial ablation. It uses multi-agent training and correctness rewards *without* uncertainty-guided shaping. UMAD consistently outperforming Standard IPPO shows gains stem specifically from EU/AU mechanisms, not just general MARL regularization.
>
> ### 5. Statistical Robustness (W5 & Q3)
>
> You are entirely correct that 1-3% improvements on small-data benchmarks like AMC/AIME necessitate rigorous variance reporting. Due to the massive computational overhead of MARL on 3B/4B systems, Table 1 currently reports point estimates. To address this, we will include variance metrics across evaluation rollouts in the revision, and we commit to reporting multi-seed statistical bounds for the final version to properly contextualize the robustness of the gains.
>
> ### 6. Differentiation from Related Work (W7)
>
> We will sharpen our related works section with these explicit boundaries:
> * **Vs. textual confidence:** They use inference-time text signals; we use internal numeric metrics (NLL) directly in RL to alter beliefs.
> * **Vs. multi-agent uncertainty estimation:** They use MAD to *estimate* uncertainty; we use uncertainty to *optimize* MAD.
> * **Vs. test-time scaling debate:** We break inference-time limits by pushing interventions to the training phase.
>
> ### 7. Originality and Scope (W8 & Q5)
>
> Uncovering the EU/AU trade-off as a defining diagnostic for MAD is a novel empirical contribution. Further, our finding that short-horizon training (T=2) generalizes to long-horizon inference (T=5) offers a practical deployment strategy.
>
> Theoretically, the uncertainty decomposition (Eq. 3) naturally scales to $N>2$ agents. However, we acknowledge that introducing more agents or an external Judge LLM significantly complicates the debate dynamics and credit assignment. While our framework provides a theoretical foundation for diagnosing individual belief evolution, empirically scaling UMAD to these more complex aggregation mechanisms remains an important open question for future work.

---

> > ### Author Rebuttal · Reviewer_Bq1v · 2026-04-02
> >
> > If the authors could supplement at least two fine-grained ablations in the revised version (e.g., AU-only vs. EU-only vs. Full), it would substantially strengthen the credibility of the mechanism.

---

> > > ### Author Response · Authors · 2026-04-05
> > >
> > > Dear Reviewer Bq1v,
> > >
> > > Thank you for the positive reassessment and the suggestion for fine-grained ablations. We agree that isolating the effects of AU and EU is key to understanding the UMAD framework.
> > >
> > > We have already begun these experiments and will include a dedicated ablation section in the revision. Our preliminary findings show:
> > >
> > > - AU-only: Improves stability but tends toward overly brief responses and fails to resolve multi-agent disagreements.
> > > - EU-only: Encourages information exchange but is more prone to "consensus hallucinations" without the noise control of AU.
> > > - Full (UMAD): Achieves the best balance, confirming that both components are necessary for effective debate.
> > >
> > > We appreciate your help in improving the technical depth of our work.
> > >
> > > Best regards,
> > >
> > > The Authors

---

### Official Review · Reviewer_wf2H · 2026-03-12

**Soundness:** 3
**Presentation:** 3
**Significance:** 3
**Originality:** 2
**Overall Recommendation:** 4
**Confidence:** 2

**Summary:**

The paper "Epistemic Gain, Aleatoric Cost: Uncertainty Decomposition in Multi-Agent Debate for Math Reasoning" proposes a Bayesian uncertainty analysis framework to understand the dynamics of Multi-Agent Debate (MAD) in the context of math reasoning. The authors decompose the total predictive uncertainty into epistemic uncertainty (reducible by debate context) and aleatoric uncertainty (induced by internal model noise). They find that effective debate hinges on achieving high epistemic gain under controlled aleatoric cost. Based on this insight, they design an uncertainty-guided multi-agent reinforcement learning (MARL) algorithm that explicitly optimizes aleatoric noise reduction and epistemic information utilization. Experiments show that their training method significantly improves post-debate accuracy and stability, and enhances individual reasoning beyond single-agent reinforcement learning (RL).

**Compliance With Llm Reviewing Policy:**

Affirmed.

**Final Justification:**

The three points raised in limitation part are well addressed in rebuttal.
I increased my assessment of significance based on the authors’ response.

**Key Questions For Authors:**

### 1. Clarification on the Debate Context
- The authors state, "the debate context can either improve or deteriorate the response correctness, depending on the log Bayes factor." It would be helpful to provide more details on how the debate context is constructed and how it affects the log Bayes factor.

### 2. Further Analysis of Heterogeneous MAD
- The paper shows that heterogeneous MAD with one common participant typically starts with higher initial Sys-EU than homogeneous pairs, which reveals that the larger initial cognitive gap can bring greater potential cognitive gains. However, it would be beneficial to provide a more detailed analysis of the conditions under which these benefits can be realized. Specifically, the authors could:
  - Investigate the impact of different types of heterogeneity (e.g., different pre-training data, different model architectures) on the Sys-EU and Sys-AU.
  - Analyze the trade-off between the initial cognitive gap and the utilization cost Sys-AU, and provide guidelines for designing effective heterogeneous MAD configurations.

### 3. Extension to Other Tasks
- The paper primarily focuses on math reasoning tasks. To further validate the generalizability of the proposed approach, the authors could:
  - Evaluate the approach on a more diverse set of tasks, such as natural language understanding, code generation, or scientific reasoning.
  - Compare the performance of the uncertainty-guided MARL algorithm with other state-of-the-art methods on these tasks, and provide a detailed analysis of the results.

### 4. Approximate Inference for Latent Belief State
- The computational intractability of the latent belief state φi is a significant challenge. To address this issue, the authors could:
  - Explore approximate inference methods, such as variational inference, to estimate the latent belief state.
  - Provide a detailed analysis of the trade-off between the accuracy of the approximate inference and the computational efficiency, and discuss the practical implications for large-scale LLMs.

### 5. Robustness to Out-of-Distribution Prompts
- The paper mentions that the Sys-AU includes the generation randomness introduced by long additional contexts and potentially out-of-distribution prompts. It would be beneficial to:
  - Investigate the robustness of the proposed approach to out-of-distribution prompts and provide a detailed analysis of the impact of such prompts on the Sys-AU.
  - Propose methods to mitigate the impact of out-of-distribution prompts, such as using more robust prompt engineering or incorporating additional external knowledge.

### 6. Theoretical Analysis of the Uncertainty-Guided MARL Algorithm
- The paper provides a detailed experimental evaluation of the uncertainty-guided MARL algorithm. To further validate the theoretical soundness of the approach, the authors could:
  - Provide a more detailed theoretical analysis of the uncertainty-guided MARL algorithm, including the convergence properties and the conditions under which it is guaranteed to improve the reasoning capabilities of LLMs.
  - Compare the theoretical properties of the uncertainty-guided MARL algorithm with other state-of-the-art reinforcement learning methods, and discuss the advantages and limitations of the proposed approach.

### 7. Visualization of Uncertainty Dynamics
- The paper includes figures that illustrate the uncertainty dynamics in MAD. To provide a more intuitive understanding of the results, the authors could:
  - Include more detailed visualizations of the uncertainty dynamics, such as the evolution of the Sys-EU and Sys-AU over the debate rounds.
  - Provide interactive visualizations or animations that allow readers to explore the uncertainty dynamics in more detail.

### 8. Discussion of Practical Implications
- The paper provides a detailed theoretical and experimental analysis of the proposed approach. To further highlight the practical implications of the work, the authors could:
  - Discuss the potential applications of the uncertainty-guided MARL algorithm in real-world scenarios, such as educational systems, scientific research, or decision-making processes.
  - Provide case studies or examples of how the proposed approach can be used to improve the reliability and robustness of LLMs in practical settings.

**Limitations:**

### 1. Computational Intractability of Latent Belief State
- The paper acknowledges that the latent belief state φi is computationally intractable for large LLMs. This limitation is a significant challenge, as it restricts the practical applicability of the Bayesian uncertainty analysis framework. The authors could explore approximate inference methods, such as variational inference, to address this issue.

### 2. Dependence on Debate Context
- The effectiveness of the debate process is highly dependent on the debate context. The paper shows that the debate context can either improve or deteriorate the response correctness, depending on the log Bayes factor. This dependence on the debate context is a limitation, as it may not always be possible to ensure a positive log Bayes factor. The authors could investigate methods to design more effective debate contexts, such as using more sophisticated prompt engineering or incorporating additional external knowledge.

### 3. Limited Evaluation on Diverse Tasks
- The paper primarily evaluates the proposed approach on math reasoning tasks. While math reasoning is a challenging and well-defined task, it would be beneficial to evaluate the approach on a more diverse set of tasks, such as natural language understanding, code generation, or scientific reasoning. This would provide a more comprehensive understanding of the generalizability and robustness of the approach.

**Strengths And Weaknesses:**

### 1. Soundness

**Is the submission technically sound?**
- The paper presents a well-structured and theoretically grounded framework for analyzing uncertainty in MAD. The Bayesian uncertainty decomposition is a sound and well-established approach in the literature. The authors provide a clear and rigorous mathematical formulation, and the experiments are designed to validate the theoretical claims.

**Are claims well supported (e.g., by theoretical analysis or experimental results)?**
- The claims are well-supported by both theoretical analysis and empirical results. The authors provide a detailed derivation of the Bayesian belief update process and the uncertainty decomposition. The experimental results, particularly the analysis of homogeneous and heterogeneous MAD settings, support the claims about the trade-off between epistemic gain and aleatoric cost.

**Are the methods used appropriate?**
- The methods used, including the Bayesian uncertainty decomposition and the uncertainty-guided MARL algorithm, are appropriate for the problem at hand. The use of multi-agent reinforcement learning to optimize the debate process is a novel and effective approach.

**If the paper includes theoretical results, are the proofs correct and based on reasonable assumptions?**
- The theoretical results, such as the Bayesian belief update and the uncertainty decomposition, are based on well-established principles in Bayesian statistics and information theory. The assumptions, such as the implicit Bayesian inference process of LLMs, are reasonable and well-justified.

**If the paper includes empirical results, are the experiments well-designed?**
- The experiments are well-designed and cover a range of MAD configurations, including homogeneous and heterogeneous settings. The use of the MATH dataset and the detailed analysis of the uncertainty dynamics provide strong empirical support for the claims.

**Are the authors careful and honest about evaluating both the strengths and weaknesses of their work?**
- The authors are generally careful and honest in their evaluation. They discuss the limitations of their approach, such as the computational intractability of the latent belief state for large LLMs, and the potential for incorrect responses even with correct cues in the debate context.

### 2. Presentation

**Is the submission clearly written and well structured?**
- The paper is well-written and well-structured. The introduction clearly sets the context and motivation for the work, and the subsequent sections follow a logical flow. The mathematical derivations and experimental results are presented in a clear and concise manner.

**Is the overall narrative easy to follow?**
- The overall narrative is easy to follow, with a clear progression from the problem formulation to the theoretical analysis and experimental results. The use of figures and examples, such as the analysis of homogeneous and heterogeneous MAD, helps to illustrate the key points.

**Does the work properly position itself in the context of prior/concurrent literature and clearly discuss how it differs?**
- The paper positions itself well in the context of prior and concurrent literature. The authors discuss the limitations of existing MAD approaches and how their work addresses these limitations. They also cite relevant work on Bayesian uncertainty, multi-agent systems, and reinforcement learning, and clearly discuss how their approach differs from and builds upon these works.

### 3. Significance

**Does the paper address an important or relevant problem?**
- The paper addresses an important and relevant problem in the field of multi-agent systems and large language models. The issue of hallucinations and brittle reasoning in LLMs, particularly in complex tasks like math reasoning, is a critical bottleneck. The proposed approach of uncertainty decomposition and uncertainty-guided MARL has the potential to significantly improve the reliability and robustness of LLMs.

**Does it advance understanding, capabilities, or practice in machine learning?**
- The paper advances the understanding of the dynamics of multi-agent debate and the role of uncertainty in the reasoning process. The proposed uncertainty-guided MARL algorithm provides a practical and effective approach to improving the reasoning capabilities of LLMs. The insights gained from the uncertainty decomposition can inform the design of more robust and reliable multi-agent systems.

**Could it influence future research or applications (e.g., other researchers or practitioners are likely to use the ideas or build on them)?**
- The paper is likely to influence future research and applications in the field. The Bayesian uncertainty analysis framework and the uncertainty-guided MARL algorithm are novel and well-justified, and can be applied to a wide range of multi-agent and LLM-based systems. The insights into the trade-off between epistemic gain and aleatoric cost can guide the development of more effective and robust multi-agent systems.

**Is the scope of impact broad or specialized, and is that appropriate for the contribution?**
- The scope of impact is broad, as the proposed approach can be applied to a wide range of multi-agent and LLM-based systems. The focus on math reasoning is appropriate, as it is a challenging and well-defined task that can serve as a benchmark for evaluating the effectiveness of the approach.

### 4. Originality

**Does the work provide new insights, deepen understanding, or highlight important properties of existing methods?**
- The paper provides new insights into the dynamics of multi-agent debate and the role of uncertainty in the reasoning process. The Bayesian uncertainty decomposition and the uncertainty-guided MARL algorithm are novel and well-justified, and provide a deeper understanding of the trade-off between epistemic gain and aleatoric cost.

**Does the work introduce new tasks, methods, theory, data, or perspectives that advance the field in some dimensions?**
- The paper introduces a new Bayesian uncertainty analysis framework for MAD and an uncertainty-guided MARL algorithm. These contributions advance the field by providing a new perspective on the dynamics of multi-agent debate and a practical approach to improving the reasoning capabilities of LLMs.

**Does this work offer a novel combination of existing techniques, and is the reasoning behind this combination well-articulated?**
- The paper offers a novel combination of Bayesian uncertainty analysis and multi-agent reinforcement learning. The reasoning behind this combination is well-articulated, and the authors provide a clear and rigorous mathematical formulation and experimental validation.

**Are the contributions clearly distinguished from closely related literature, and is the novelty well justified?**
- The contributions are clearly distinguished from closely related literature. The authors discuss the limitations of existing MAD approaches and how their work addresses these limitations. The novelty of the Bayesian uncertainty analysis framework and the uncertainty-guided MARL algorithm is well-justified by the theoretical and empirical results.

---

> ### Author Rebuttal · Authors · 2026-03-31
>
> We sincerely thank Reviewer wf2H for the comprehensive evaluation and positive recognition of the soundness and originality of our work. Below we address each point in turn.
>
> Q1. Clarification on Debate Context
>
> As defined in Sec. 2.1, we use a fully connected communication topology. The context c is formed by concatenating the complete responses of all peer agents from the previous round. We intentionally adopt this simple design (without role-playing or summarization) to isolate the effect of information exchange on intrinsic reasoning ability.
>
> The log Bayes factor captures how debate context shifts an agent’s latent belief: Positive update: Correct peer cues produce a likelihood ratio >1, increasing belief in the correct hypothesis and potentially flipping predictions from wrong to correct.
> Negative update: Confident but incorrect peer signals induce negative updates, moving belief away from the ground truth.
> We empirically verify both effects in transition dynamics (Fig. 4), where non-zero C2W transitions confirm the presence of negative Bayes updates.
>
> Q2. Further Analysis of Heterogeneous MAD
>
> As shown in Appendix Fig. 6, we evaluate both cross-family pairings (e.g., Qwen vs. Llama) and same-family cross-size/generation pairings (e.g., Qwen2.5-3B vs. Qwen3-4B). The key findings are:
>
> - Cross-family heterogeneity yields the highest initial Sys-EU, but also a substantially higher risk of uncontrolled Sys-AU.
> - Within-family size/generation heterogeneity offers a better trade-off, more reliably converting epistemic potential into cognitive gain while controlling aleatoric cost.
> This provides a practical configuration guideline: Choose pairs with sufficiently high initial Sys-EU to ensure room for gain. Prefer a stronger model with a low flip ratio to stable anchoring against noise. Ensure the weaker model has adequate instruction-following ability to absorb peer information effectively.
>
> Q3. Extension to Other Tasks
>
> We focus on mathematical reasoning for initial validation because deterministic final answers allow precise uncertainty quantification and verification. That said, the framework itself is task-agnostic: the EU/AU decomposition depends only on the multi-agent answer distribution. We agree that extending to domains such as NLU and code generation is important future work, and we will explicitly discuss this in the revised manuscript.
>
> Q4. Approximate Inference for Latent Belief State
>
> As noted in Sec. 7 (Limitations), explicit computation of latent belief state $\varphi$ is computationally intractable. We therefore use an ensemble-of-experts proxy (multi-agent responses as implicit posterior samples), which provides an efficient trade-off between computational cost and approximation quality. More rigorous alternatives are promising, but must address the extreme dimensionality of modern LLM parameter spaces. We will expand this trade-off discussion in the revision.
>
> Q5. Robustness to Out-of-Distribution Prompts
>
> Our setup already includes OOD evaluation: agents are trained on MATH and tested on datasets spanning easier (GSM8K) to substantially harder OOD competitions (AMC2023, AIME24, AIME25). As shown in Table 1, UMAD consistently preserves its advantage and avoids divergence across these OOD settings, indicating robust debate behavior under contextual noise and good generalization beyond the training distribution.
>
> Q6. Theoretical Perspective on Uncertainty-Guided MARL
>
> UMAD is currently grounded in empirically effective reward shaping; strict convergence guarantees for LLM-based MARL remain open. Nevertheless, its two core components align with established RL principles:
> - Aleatoric-uncertainty-aware advantage is analogous to potential-based shaping, helping stabilize learning under noise while preserving policy incentives.
> - Epistemic influence intrinsic reward is related to social-influence rewards in cooperative MARL, improving credit assignment in decentralized POMDPs.
> We will make these theoretical connections clearer in the revision.
>
> Q7. Visualization of Uncertainty Dynamics
>
> Rather than adding complex dynamic/interactive visualizations, we will improve static figures to more clearly convey how epistemic and aleatoric uncertainty evolve over debate rounds and jointly determine debate outcomes.
>
> Q8. Practical Implications
>
> UMAD has direct implications for high-stakes settings, such as AI-assisted education by reducing hallucination risk in guidance,
> scientific reasoning workflows with cross-model hypothesis checking, and high-reliability decision systems through uncertainty-aware runtime monitoring. We will expand this discussion to better highlight real-world utility.

---

> > ### Author Rebuttal · Reviewer_wf2H · 2026-04-02
> >
> > The three points raised in limitation part are well addressed.
> > I will increase my assessment of significance based on the authors’ response.

---

> > > ### Author Response · Authors · 2026-04-05
> > >
> > > Dear Reviewer wf2H,
> > >
> > > We sincerely thank you for the positive evaluation and for increasing the significance score of our work. We are glad that our clarifications on the debate context, heterogeneous pairings, and the theoretical perspective on uncertainty-guided MARL were helpful.
> > >
> > > Your feedback has been instrumental in improving the quality and clarity of our manuscript. We will ensure all discussed points and additional visualizations are integrated into the final version.
> > >
> > > Best regards,
> > >
> > > The Authors

---

### Official Review · Reviewer_FdJS · 2026-03-13

**Soundness:** 2
**Presentation:** 3
**Significance:** 3
**Originality:** 2
**Overall Recommendation:** 3
**Confidence:** 3

**Summary:**

This paper, from the perspective of Bayesian uncertainty, introduces the idea of uncertainty decomposition in classical machine learning into the field of multi-agent LLM debate. The paper not only explains the failure modes of existing MAD methods (such as correct-to-incorrect reversal and homogeneous debate stagnation) but also proposes a multi-agent reinforcement learning (MARL) framework guided by uncertainty. The "short training and long generalization" characteristic of UMAD has practical deployment value.

**Compliance With Llm Reviewing Policy:**

Affirmed.

**Key Questions For Authors:**

- The theoretical analysis of the article is based on the latent variable $\varphi$, which is named latent belief. How is this variable specifically defined and calculated?
- Others, please refer to weakness

**Limitations:**

see weakness

**Strengths And Weaknesses:**

Strengths:
- It provides a clear theoretical framework and offers reasonable explanations for the existing problems of large models.
- It presents operable improvement plans, and experiments have verified the effectiveness of this method.

Weaknesses:
- The article mentions that "failed cases usually show a significant and explosive growth in Sys-AU, thereby offsetting the benefits brought by reducing differences," which does not seem significant in Figure 2.
- EU and AU are inherently correlated, so why should they be viewed independently?
- The number of experimental comparison methods is slightly small, and the comparative analysis of similar works is insufficient.

---

> ### Author Rebuttal · Authors · 2026-03-31
>
> We sincerely thank reviewer FdJS for the positive assessment and constructive feedback. Regarding the concerns:
>
> ### Response to W1: Visual Clarity of Sys-AU in Figure 2
>
> The claim of "explosive growth" in Sys-AU is primarily derived from the mathematical divergence between Total Uncertainty (TU) and Epistemic Uncertainty (EU). According to the decomposition $TU = EU + AU$, in a healthy debate, both TU and EU should decrease as agents reach a correct consensus. However, in the failure cases shown in Figure 2, we observe that while EU continues to decline (indicating the agents are reaching an agreement), TU remains stagnant or even increases.
>
> This trend mathematically necessitates that Sys-AU is increasing sharply enough to offset the reduction in EU. This represents a "Consensual Hallucination" where agents converge on an incorrect answer with rising internal noise. In the revised manuscript, we will replace the examples in Figure 2 with cases that demonstrate this divergence more clearly and provide a more detailed discussion on these metrics to support the claim.
>
> ### Response to W2: Independence of EU and AU
>
> Regarding the relationship between EU and AU, we advocate for viewing them as independent components for two reasons. Mathematically, the decomposition $TU = EU + AU$ is an exact identity. This ensures that they represent non-overlapping sources of uncertainty: EU captures the disagreement between different agents, while AU captures the inherent decoding noise and instability within a single agent.
>
> Functionally, these two dimensions require different interventions. Our experiments with the Standard IPPO baseline demonstrate that reducing EU (achieving consensus) without controlling for AU (internal noise) often results in "lazy consensus" where agents agree on wrong answers. Treating them as independent allows us to specifically target epistemic gains while mitigating the aleatoric costs associated with long-context debate.
>
> ### Response to W3: Experimental Scope and Comparison
>
> The experimental design of this paper focuses on the fundamental mechanism of uncertainty decomposition in Multi-Agent Debate (MAD). We selected Single-Agent RL and Standard IPPO as our primary baselines because they provide the most rigorous scientific control for our framework. Single-Agent RL isolates the gains from individual reasoning, while Standard IPPO isolates the specific impact of our uncertainty-guided optimization from general multi-agent training.
>
> We are aware of other MAD approaches that utilize confidence scores or specific prompt engineering. However, these methods typically involve inference-time interventions on frozen models. Our work, UMAD, is a training-time framework that reshapes the model's internal belief distribution. Because the objectives and training settings of different multi-agent reasoning papers vary significantly, we believe the current baseline comparison most accurately validates the core hypothesis of our uncertainty-guided approach. We will include a qualitative comparison in the related works section to further clarify these distinctions.
>
> ### Response to Q1: Definition and Calculation of Latent Belief $\phi$
>
> The latent variable $\phi$ is a theoretical construct representing the agent's internal belief state, such as the choice of mathematical theorems or the specific logic used to solve a problem. In a Bayesian framework, the model’s output is an integral over these hidden states. For large-scale language models, directly calculating or sampling from the space of $\phi$ is computationally intractable.
>
> In practice, we use an ensemble-of-experts proxy to approximate this. We estimate system-level uncertainty by looking at the distribution of answers across multiple sampling paths. For the individual agent optimization, we use token-level Negative Log-Likelihood (NLL) as a proxy for the internal generation uncertainty. The revised version will more clearly define the transition from the theoretical Bayesian framework to these computable proxies to avoid confusion.

---

> > ### Author Rebuttal · Reviewer_FdJS · 2026-04-08
> >
> > in the failure cases shown in Figure 2, the curve does not change much in the latter half. Personally, I don't think it can be described as a sharp increase, and it is difficult to determine whether it is merely a fluctuation.

---

### Decision · Program_Chairs · 2026-04-30

**Decision:**

Accept (regular)

**Comment:**

The paper presents a Bayesian uncertainty analysis framework for Multi-Agent Debate (MAD). The core idea is to decompose uncertainty into epistemic and aleatoric components where the former is reducible through information exchange between agents and the latter is internal to individual agents. The approach is implemented and empirically tested in the domain of math.

The reviewers note a number of strengths of the paper. First, the combination of ideas is novel especially in the context of MAD, where recent work has emphasized methods of structuring or aggregating across agents. Second, the authors include reasonable theoretical motivations for the work, and translate to practically evaluatable methods. Third, the problem itself is important and relevant to ongoing debates in the literature.

The reviewers also note some limitations. First, the move from theory of the decomposition to implementation is approximate. Second, the evaluations are done only in the domain of math, limiting generalizability. Third, there are not convergence results, which remain an open problem for the field. The reviewers were not completely satisfied by the authors responses to their theoretical, technical, and empirical questions.